



# On the validation of K index values at Italian geomagnetic observatories

Mauro Regi[1], Paolo Bagiacchi[2], Domenico Di Mauro[2], Stefania Lepidi[1], Lili Cafarella[2]

[1]Istituto Nazionale di Geofisica e Vulcanologia, 67100, L'Aquila, Italy
[2]Istituto Nazionale di Geofisica e Vulcanologia, 00143, Rome, Italy

*Correspondence to*: Mauro Regi (mauro.regi@ingv.it)

**Abstract.** Local K index and the consequent global Kp index are well established three-hour range indices used to characterize the geomagnetic activity. K index is one of the parameters which INTERMAGNET observatories can provide and it's widely used since several decades, although many other activity indices have been proposed in the meanwhile. The

method for determining the K values should be the same for all observatories; so INTERMAGNET consortium recommends a software code, KASM designed for an automatic calculation of K index according to the Adaptive Smoothed method. K values should be independent on the local dynamic response, therefore for their determination each observatory has its own specific scale regulated by the K9 lower limit, which represents the main input parameter for KASM. The determination of an appropriate K9 value for any geomagnetic observatory is then fundamental. In this work we statistically analyze the K

values estimated by means of KASM code for the Italian geomagnetic observatories of Duronia (corrected geomagnetic latitude λ~36° N) and Lampedusa (λ~28° N) comparing them with the German observatories of Wingst and Niemegk. Our comparative analysis is finalized to establish the best estimation of the K9 lower limit for these two stations. A comparison of K9 lower limits found for the Italian observatories with results from a previous empirical method was also applied and used to verify the consistency and reliability of our outcomes.

## 1 Introduction

In their pioneering work, Bartels et al. (1939) introduced the three-hour-range *K* index with the purpose of quantifying the solar wind (or particle) effects on the geomagnetic field. K index is represented with an integer in the range 0–9 ("K" is from the German word Kennziffer meaning "characteristic digit") with 0 and 1 being an indication of quiet condition and 5 or

more referring to an increased level of magnetic activity, generally related to a geomagnetic storm. It is derived for a specific observatory from the maximum fluctuations of horizontal components observed on a magnetogram during a three-hour interval, evaluated as difference between maximum positive and negative deviations with respect to a reference curve which essentially reflects the local diurnal variation at the observatory. These maximum deviations may occur at any time during the 3 hour period. The proposed K index was originally calculated for Niemegk observatory.

As a natural consequence of K index, the planetary geomagnetic activity index Kp was proposed by Bartels (1949). It is derived from the standardized K index (Ks) of 13 magnetic observatories at mid latitude and it is representative of the large



spatial-scale of the solar wind-magnetosphere coupling energy. Therefore, K index is the fundamental parameter for Kp estimation that is widely used, for example, in space weather applications, for identify quietest days (Johnston, 1943) used also in the IGRF modeling, for verifying solar wind driven modulation in the atmospheric parameters during disturbed
conditions (Regi et al., 2017).

The main difficulty for K indices evaluation is to assign a proper quasi-logarithmic scale to the geomagnetic fluctuations that satisfy the principle of the assimilation of frequency distributions (AFD): the frequency distributions (or occurrences) of K index values at different sites are, in principle, the same (Bartels et al., 1939). In other words, *A* values vary from observatory to observatory in such a way that the historical rate of occurrence of certain levels of K is about the same at all
observatories (Bartels-Mayaud rules). This implies that, for a given K value, $A_K$ increases with increasing latitude, and the fundamental quantity for the K index calculation is represented by the minimum amplitude K9 corresponding to K=9, from which also the other $A_K$ values are derived. For example (see Bartels et al., 1939), from higher to lower latitude, at Sitka (AACGM latitude $\lambda$=52° N, Alaska), K9=1000 nT, while in Honolulu ($\lambda$=21.37° N, Hawaii) K9=300 nT. The GFZ website (https://www.gfz-potsdam.de/en/kp-index/) provides K9 values for the 13 observatories used for Kp evaluation, showing
values between 450 nT and 1500 nT; in particular, at Niemegk ($\lambda$=47.94° N, Germany) K9=500 nT.

For many years, K was manually derived by means of a conversion table containing the values of the maximum fluctuation *A*, expressed in units of nT, for each K value. With the introduction of digitized data and with the increasing access to computers, the manual estimation of K index was progressively substituted with automated algorithms and, nowadays, the reproducibility, one of the cornerstones of science, has become possible. This implies the production of computer plots of
digital data with scale values similar to those of photographic magnetograms (Menvielle et al., 1995). The International Association of Geomagnetism and Aeronomy (IAGA, http://www.iaga-aiga.org/) promotes tools or methods able to make it easier to keep track of files and analyses done on computers.

Different methods were proposed and carefully compared and assessed in occasion of an international meetings organized by the IAGA Working Group "Geomagnetic indices" during the Vienna IUGG general Assembly in 1991 and four methods
were acknowledged: FMI (provided by Finnish Meteorolical Institute, Finland), Hermanus (Hermanus Magnetic Observatory, CISR, South Africa), KASM (Institute of Geophysics, Polish Academy of Science) and USGS (USGS, USA), whose Fortran 77 codes are available at the International Service of Geomagnetic Indices (ISGI, http://isgi.unistra.fr/softwares.php).

The International Real-time Magnetic Observatory Network (INTERMAGNET, http://www.intermagnet.org), of which
IAGA is associated, endorses and recommends KASM for calculation of geomagnetic activity indices K according to the Adaptive Smoothed method (Nowozyński et al., 1991). For the calculation of the K index, IAGA formatted files are used by KASM code. It requires three daily files, the one under analysis and the files of the previous and following days on which





the code derives daily values without fluctuations (mainly daily variation). The code also needs as input parameters the K9 value and the yearly average of the H component relative to the year of interest.

We want to point out that it does not exist an unique K9 at a given geomagnetic latitude since the geomagnetic activity shows a well known magnetic local time (MLT) dependency and, in addition, each site could be affected by different local features such as, for example, crustal anomalies (Chiappini et al., 2000) and/or coast effect (Parkinson, 1962; Regi et al., 2018). For the inclusion of a new geomagnetic observatory into the INTERMAGNET network, K9 should be assigned, for example, by comparing geomagnetic field variations between the new observatory and the historical ones for which K

indices are estimated by using well defined K9 levels, obtained from a long time observation.

We used the geomagnetic data from the two Italian geomagnetic observatories at Duronia (DUR) and Lampedusa (LMP), evaluating the K index with the purpose of estimating the best K9 value for each observatory.

DUR observatory is operating in Central Italy in the area of the village of Duronia (geogr. coordinates: 41°39'N, 14°28'E, 910 m a.s.l.). It was installed at the end of 2007 in the framework of MEM (Magnetic and Electric fields Monitoring) Project

that aims to investigate the environmental electromagnetic signals in the ULF-VLF (0.001 Hz - 100 kHz) frequency band, and was granted as geomagnetic observatory in 2012, when it was included in the INTERMAGNET network (http://www.intermagnet.org), replacing the historical geomagnetic observatory at L'Aquila partially damaged after the local $M_w$ 6.2 earthquake in 2009.

LMP is the southernmost observatory in Europe (geogr. coordinates: 35°31'N, 12°32'E); it was installed in 2005 and is

regularly working since 2007.

Up to now, K index was evaluated only for DUR observatory, using K9=350 nT, both for hand-scaling (since 2012) and for KASM program (since 2017).

In this work we evaluated K9 throughout a correlation analysis performed between *K* index at DUR with that provided by historical observatories. In order to take into account the magnetic local time dependency of reference *K* index, European

observatories were selected. As possible reference observatories we chose Wingst (WNG) and Niemegk (NGK), since they are among the 13 observatories that contribute to the Kp estimation and their local magnetic time is quite close to that of our Italian observatories.

Our investigations suggest that NGK is the best reference observatory for Italian geomagnetic observatory of DUR, probably due to the closest magnetic local times: by comparing DUR with NGK we estimated a reliable DUR K9 level of 320 nT.

Finally, by comparing also LMP with NGK, a reliable LMP K9 level of 310 nT is estimated.





## 2 Data and methods of analysis

Geomagnetic field variations at Italian geomagnetic observatories of DUR and LMP are measured by using three-axis
fluxgate magnetometers along the northward (*H*), eastward (*D*), and vertically downward (*Z*) directions in the geomagnetic
reference frame at 1 s sampling rate. Following the INTERMAGNET directives, geomagnetic time series are also stored as
daily archives at 1 min sampling rate, according to the IAGA 2002 format.

In this work we used data available in the time interval 1 January 2017 – 31 December 2018, a temporal window which falls
in the lower part of the sunspot number curve for the cycle 24 (Upton & Hathaway, 2018).

These data are used for estimating K indices by using KASM algorithm which is recommended by INTERMAGNET. In this
work, the definitive K9 level at DUR is empirically estimated throughout the following procedure:

    a)   we selected a reference observatory;

    b)   K index time series at DUR are computed by using KASM for different K9 values( $K_{K9}$ ) in the range 200-400 nT
        with a step size of 10 nT;

c)   each $K_{K9}$ index time series at DUR is compared with K index time series at reference observatory through
        correlation analysis;

    d)   the definitive K9 level at DUR is estimated in correspondence of the maximum correlation coefficient.

As possible reference we selected the historical observatories of NGK and WNG since they are among the 13 observatories
used for Kp evaluation, they are both in Europe at a MLT close to that of DUR and LMP (see Tab.1 and Fig.1), which is at
the moment the principal Italian observatory, member of the INTERMAGNET network and then used in this work as
reference observatory for any Italian site. By following our procedure at point c), using independently NGK and WNG, we
found that the higher correlation is reached with NGK. The same procedure from a) to d) is applied to the lower latitude
Italian observatory of LMP.

We note that K indices at NGK and WNG are both generated by using FMI algorithm. Then, we find useful to verify that
FMI and KASM are consistent methods by comparing K values estimated with both methods at NGK.

Finally, we compared K9 values estimated at Italian geomagnetic observatories by means of our method, with those
estimated using an historical method proposed by Mayaud (1980).




# 3 Experimental results

## 3.1 K9 empirical estimation

We retain that it is important to know how K indices are distributed at consolidate reference observatories of NGK and WNG and how they are in relation to each other.

Figure 2 shows the K index at NKG (panel a) and WNG (panel b) and the difference $\Delta K = K_{NGK} - K_{WNG}$ (panel c) during 2017-2018; it also shows (panel d) the K index frequency distributions $v$ (or occurrences) at the two observatories. We can see that the two frequency distributions are very close, as confirmed by the distribution of $\Delta K$ (panel f), with the largest number of cases in correspondence of $\Delta K = 0$ (4680 cases, ~80%), and by the absence of cases with $|\Delta K| > 1$. However, $\Delta K$ distribution shows also a non-symmetric distribution around zero, with a very different number of cases in correspondence of $\Delta K = \pm 1$:

1103 cases (~19%) for $\Delta K = -1$ and 48 cases (~1%) for $\Delta K = +1$; this feature is also evidenced by the linear regression law: $K_{NGK} = \alpha K_{WNG} + \beta$ (panel e), where by imposing $\beta = 0$ it is obtained $\alpha = 0.914 \pm 0.004$, i.e. $\alpha < 1$.

We investigated the frequency distribution $v$ of K at NGK in correspondence of $\Delta K = \pm 1$ cases, and compared these distributions with the general distribution of K at NGK (the one shown in Fig.2d). Figure 3 shows these distributions, separately for $\Delta K = -1$ and $\Delta K = +1$ conditions during 2017-2018 (top panels), and separately for the two years (middle and

bottom panels). In each panel, the K occurrences at NGK, regardless of $\Delta K$, are superimposed (red lines). It can be seen that the general distributions of K and $\Delta K = -1$ are more similar than those of K and $\Delta K = +1$. Therefore, the occurrences for $\Delta K \neq 0$ are not led by particular magnetospheric activity conditions.

Figure 4 shows the result of the correlation analysis between K indices at Italian observatories of DUR (panel a) and LMP (panel b) with those at NGK (red thin line) and WNG (black thin line) as functions of K9 level used by KASM for the time

interval 2017-2018. In this figure K9 levels are in the range 200-450 nT, with a step size of 10 nT. The tick lines, which show the smoothed curves computed by using a five-points triangular window, will be used hereafter as actual experimental results for our investigations. It can be seen that the correlation $r$ is higher for DUR-NGK observatories ($r \sim 0.915$ for K9=320 nT) with respect to DUR-WNG observatories ($r \sim 0.908$ for K9=290 nT). Regarding LMP, the correlation attains lower values with respect to DUR, with maximum values of $r \sim 0.875$ for K9=310 nT with NGK, and $r \sim 0.870$ for K9=300 nT with WNG.

As expected, both the K9 limit and $r$ increase with the increasing geomagnetic latitude of referred observatory, here represented by DUR and LMP. In addition, the higher correlations are obtained by using NGK, probably due to the lower latitude (i.e. closer to the Italian observatories) and the closer MLT with respect to DUR (table 1). Also at LMP, even if the MLT is closest to that of WNG, the higher correlation is found with NGK: this result suggests that latitudinal effects are dominant with respect to MLT ones. This can be well understood taking into account that the MLT range of all selected

observatories is within 11 minutes, well shorter than the 3-hour interval used for K determination.





Anyway, from these results we can assert that for the comparison with Italian observatories NGK is slightly better than WNG, and it will be used hereafter as the reference observatory for DUR and LMP. We can also assume that the best estimation of the K9 value at DUR and LMP is 320 nT and 310 nT, respectively; so the K indices computed by using KASM at DUR with K9=320 nT and at LMP with K9=310 nT represent the best input parameter for the K evaluation for Italian

observatories.

Figure 5 shows the frequency distribution of the difference between these K-index time series and that computed at NGK. The occurrences for both DUR and LMP are distributed around zero and in the range [-1:+1]. For a comparison, we show also the difference distribution obtained for DUR using K9=350 nT (dashed blues line); we recall that this is the K9 value we used up to now. It can be seen that, while for K9=320 nT the distribution is almost symmetric around zero, using K9=350 nT

it appears more asymmetric, unbalanced towards positive values, confirming that a higher correlation between NGK and DUR is found for K9=320 nT. At LMP the distribution appears slightly asymmetric, and this discrepancy with respect to DUR could be attributed to the larger latitudinal and MLT difference between NGK and LMP.

Since our validation procedure aims to estimate comparable K indices at Italian observatories, we found useful to compute ΔK between DUR and LMP, whose distribution is shown in Fig. 6. It is almost symmetric around zero, closely reflecting the

distribution of ΔK computed between K indices at LMP and NGK (from Fig.5); we can also see that |ΔK| never exceeds 1, confirming the validity of the results obtained at DUR and LMP by using KASM.

It should be noted that our comparative investigation is based on K indices at reference observatories of NGK and WNG, which are computed by using FMI algorithm with K9=500 nT, while at DUR and LMP K indices are estimated by using KASM. Therefore, the question arising from our calibration method is: are FMI and KASM algorithms consistent?

In order to answer to this question, we performed a correlation analysis between K indices obtained by using FMI ($K_{FMI}$) and KASM ($K_{KASM}$), where the latter K index is obtained for different K9 levels. In this regards we used NGK 1-min data, provided by the INTERMAGNET web site. Since at the moment NGK geomagnetic field measurements are stored as definitive and quasi-definitive data for 2017 and 2018 respectively, we preferred to separate the analysis for the two years.

Figure 7a shows the correlation analyses between K indices at NGK (from FMI algorithm) and that computed by KASM by

using K9 in the range 350-600 nT, with a step size of 10 nT. It can be seen that, the *r* maxima (~0.96 and ~0.95) are reached for K9=460 nT in both years. Assuming that $K_{KASM}$ computed for K9=460 nT represents the better K indices in comparison with $K_{FMI}$, we examined the occurrences of $ΔK=K_{FMI}-K_{KASM}$ (Fig.7, panels b and d). We can see that for ~90% of cases (and for both years) ΔK is equal to zero. Frequency distributions of $K_{KASM}$ (460 nT) and $K_{FMI}$ (500 nT) indices for the years 2017 (panel c) and 2018 (panel e) are also shown. We point out how the distributions are close to each other, suggesting that FMI

and KASM are consistent algorithms, even if they are based on different K9 limits applied for the same observatory.





## 3.2 Comparison with a previous K9 estimation method

As explained in the introduction, the geomagnetic indices are historically assigned throughout visual inspection of magnetograms. The main difficulty for K indices evaluation is to assign a proper value for the K9 limit from which determining the quasi-logarithmic scale to the geomagnetic fluctuations in order to satisfy the AFD principle (Bartels et al., 1939). Mayaud (1980) proposed a method for calculating the geomagnetic activity level $L$ at a given site by comparing the amplitude of geomagnetic fluctuations at the reference observatory ($A_0$) with that, for example, at new one ($A$) as follows: $L = L_0 A/A_0$, where $L_0$ represents the level of geomagnetic activity at the reference observatory, equivalent to K9, and all quantities are dependent on $\delta = \min[ \lambda_{oval} - \lambda ]$, i.e. the minimum angular separation between the site, located at geomagnetic latitude $\lambda$, and the auroral region, at $\lambda_{oval}$. According to Mayaud (1980), an approximate value of $\delta$ could be given by $\delta = 69° - \lambda$ but this is really just a rough approximation.

We searched a simple relationship which relates K9 (or $L$) to the geomagnetic latitude of the observatory.

As showed by Mayaud (1980), K9 increases with decreasing $\delta$ (K9$\propto\delta^{-1}$), as expected for a geomagnetic field induced by a current system. Figure 8 shows K9($\delta$) (blue points) provided in Tab. 5 by Mayaud (1980), considering only northern hemisphere. These points are well represented by a linear law considering an increasing induction effect with increasing parameter $x = 1/\delta$.

Therefore, by using $x = 1/\delta$, previous relationship is linearized and can be formulated as follows

$$K9(x) = \alpha x + \beta ,\tag{1}$$

The results of the linear regression analysis performed on the experimental points are also reported in Fig.8.

Equation 1 is therefore useful for estimating a reasonable K9 limit at a different site. In order to evaluate K9 at DUR, LMP and, for a comparison, at NGK observatories, the corresponding $\delta$ parameter is required. However, it is not clear how $\delta$ was estimated by Mayaud (1980), since it requires, for example, an auroral oval model for estimating the $\lambda_{oval}$, and an IGRF model for evaluating the geomagnetic latitude $\lambda$ of a given site (this aspect will be further discussed at the end of this section).

In this regard, we empirically estimated $\delta(\lambda)$ by a linear fit of the experimental data reported by Mayaud (1980). Figure 9 shows experimental points (red stars) and the corresponding linear law (blue line)

$$\delta(\lambda) = a\lambda + b ,\tag{2}$$





which allows us to extrapolate an estimation of the theoretical $\delta_{th}(\lambda)$ for the observatories of DUR, LMP and, for comparison at NGK too, where $\lambda$ represents the corrected geomagnetic latitude. Finally, by inserting $\delta_{th}$ into the Eq. (1) we estimated the K9($\delta_{th}$) level at the observatories.

All these results are reported in Tab.2, which also shows for a comparison, K9 obtained by computing $\delta_a=69°-\lambda$, and K9$_{exp}$ experimentally derived by our calibration procedure (K9$_{exp}$), together with the 95% confidence intervals for the fitted K9 values.

It can be seen that all K9$_{exp}$ are consistent to each other within their respective confidence interval, at a given observatory. The small difference between K9$_{exp}$ and K9$_{th}$ could be due to a different method for calculating the geomagnetic coordinates
used in Mayaud and in this work (we use AACGM). In order to verify this hypothesis, we performed a correction on the key parameter $\delta(\lambda)$ as follows:

we computed the AACGM latitudes $\Lambda$ of geomagnetic observatories from Tab. 5 of Mayaud and corrected $\lambda$ through linear relationship $\lambda_C = l\, \Lambda + m$;

we performed a linear fit of $\delta(\lambda)$, $\lambda_C$, which provides the relationship for the adjusted $\delta_A(\lambda_C)$;

finally we performed a linear fit of K9($\delta$), $\delta_A$ which provides the adjusted K9$_A(\Lambda)$ estimated at our geomagnetic observatories as a function of AACGM latitude.

With respect to the K9($\delta$) it can be seen that the adjusted K9$_A(\Lambda)$ (shown in Tab.2) are closer to the experimental K9$_{exp}$, indicating that the correction on geomagnetic coordinate makes a significant contribution on the K9 estimation. LMP is the only one that shows a discrepancy between K9$_{exp}$ and K9 here estimated with different methods. A possible reason of this
discrepancy lies in the low latitude of LMP observatory where the ring current and/or electrojet currents dynamics could affect K9 estimations.

## 4 Discussion and Conclusions

The modern automatic procedures for calculating local K index values, with the setting of some a-priori criteria, have to be
carefully verified for their permanent validation in terms of accuracy and stability when delivered to the scientific community. Automatic computation of the local K index for the Italian geomagnetic observatories of Duronia (DUR) and Lampedusa (LMP) is carried out through the KASM code. This code is distributed by the INTERMAGNET consortium and endorsed by the IAGA organization, but other codes, including the FMI, a further code considered in this paper, are in use at other historical observatories. An input parameter required by KASM code, as well as FMI code, is the K9 value, which





represents the minimum value of the amplitude extent in the H component of Earth's magnetic field when the local K value
reaches the integer 9, the highest level in a scale which ranges from 0 to 9.

We found K9 values for DUR and LMP through a correlation analysis using as reference the corresponding data from the
two European observatories of Wingst (WNG) and Niemegk (NGK), both located in Germany. The choice of these two
observatories was prompted by the fact that they are among the 13 observatories which provide their K indices for the
determination of the planetary Kp index and moreover, their magnetic local time is very close to that of the Italian
observatories. Based on a dataset related to a couple of years (2017 and 2018), this analysis allowed to establish that for
DUR and LMP the K9 values are 320 nT and 310 nT, respectively. The method can be generalized and applied to every
observatory in the world to verify if the choice to scale local fluctuations of the Earth's magnetic field is properly calibrated
by a suitably selected K9 value, regardless if manually or automatically computed. Our analysis also highlighted the
possibility of establishing a linear relationship between a pair of analyzed observatory datasets that can be useful for
predicting or deriving the index of one when the other is known.

Another interesting result that we found is related to the consistency of the KASM code and the FMI code, the latter in use at
the two German observatories for the K index computation and subsequent release. Although FMI code is based on a
different procedure, we verified that the results obtained are consistent with those obtained by KASM code and stable in the
two-year time interval, although with a slightly different value of the input K9 parameter. This confirms that the choice of a
certain algorithm in place of another does not invalidate the results.

Before the introduction of automatic procedures, based on the definition introduced by Bartels et al. (1939) for the K index
concept, in the '80s of the last century Mayaud (1980) used an empirical relation to calculate the level of the local magnetic
activity L (equivalent to the K9 values) for a generic point of observation with respect to a referenced observatory. Through
a linearization process, we used this relation, which includes some approximations and the necessity of determining the
minimum angular separation between the observational point and the auroral region, i.e. a method for determining the
geomagnetic latitude, obtaining an independent estimate of the K9 values for our observatories which is consistent, within
the 95% interval of confidence, with that obtained by our previous analysis. Moreover, Mayaud (1980) note that the
limitation of the method they propose is that it is conceived for sub-auroral and mid latitudes; indeed, they suggest that for
lower latitudes a constant K9=300 nT can be chosen. This very approximate value is not very far from the values we
estimate (320 nT for DUR and 310 nT for LMP), but would certainly be not accurate as them in the comparison with the
values from other reference observatories: indeed our results clearly show that a very precise K9 limit is necessary for
obtaining K values well consistent at different sites. As a final remark, from the overall view of this work, we are also
definitely convinced that the habit to round the value of K9 in multiples of 50 nT is a simplified approximation, firstly
suggested by Bartels et al. (1939), a practice that needs to be abandoned. This approximation is still adopted in some cases,
demonstrating that perhaps a critical revision has not been applied yet, differently from the case of Kakioka observatory
(Japan) where K9 has a convincing value of 296 nT.





## Author contribution

DDM, SL and MR planned the study. MR performed the data analysis and wrote the codes. PB studied the KASM code functionality and, together with MR, tested and validated its results. DDM, LC and SL improved the quality of the manuscript. All authors read and approved the final manuscript.

## Competing interests

The authors declare the they have no competing interests.

## Acknowledgments

The results presented in this paper rely on data collected at magnetic observatories. We thank the national institutes that support them and INTERMAGNET for promoting high standards of magnetic observatory practice (www.intermagnet.org).

We also thank Jürgen Matzka from Helmholtz Centre Potsdam GFZ German Research Centre for Geosciences (Germany) for providing K indices at Niemegk and Wingst.

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





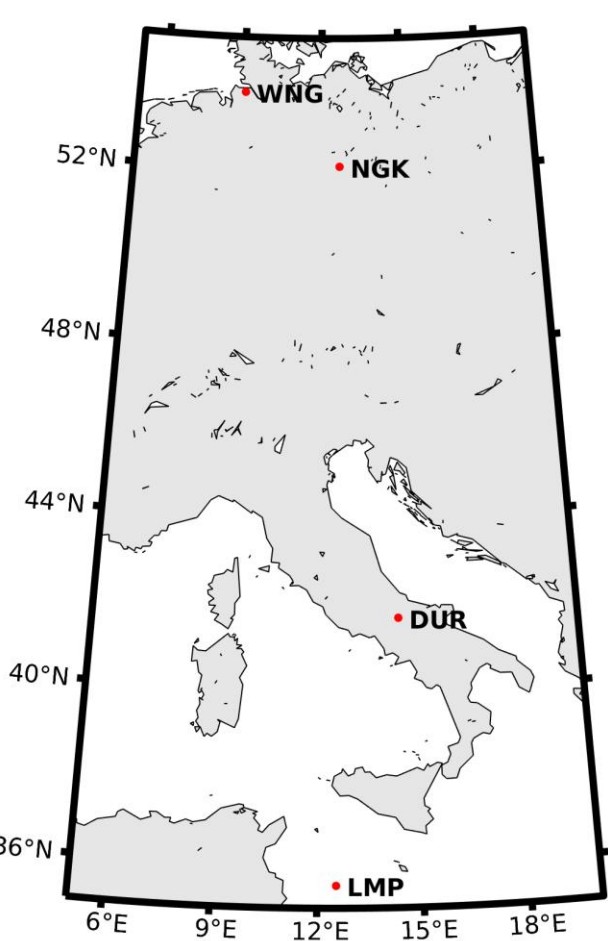

**Figure 1:** European geomagnetic observatories used in this work



340

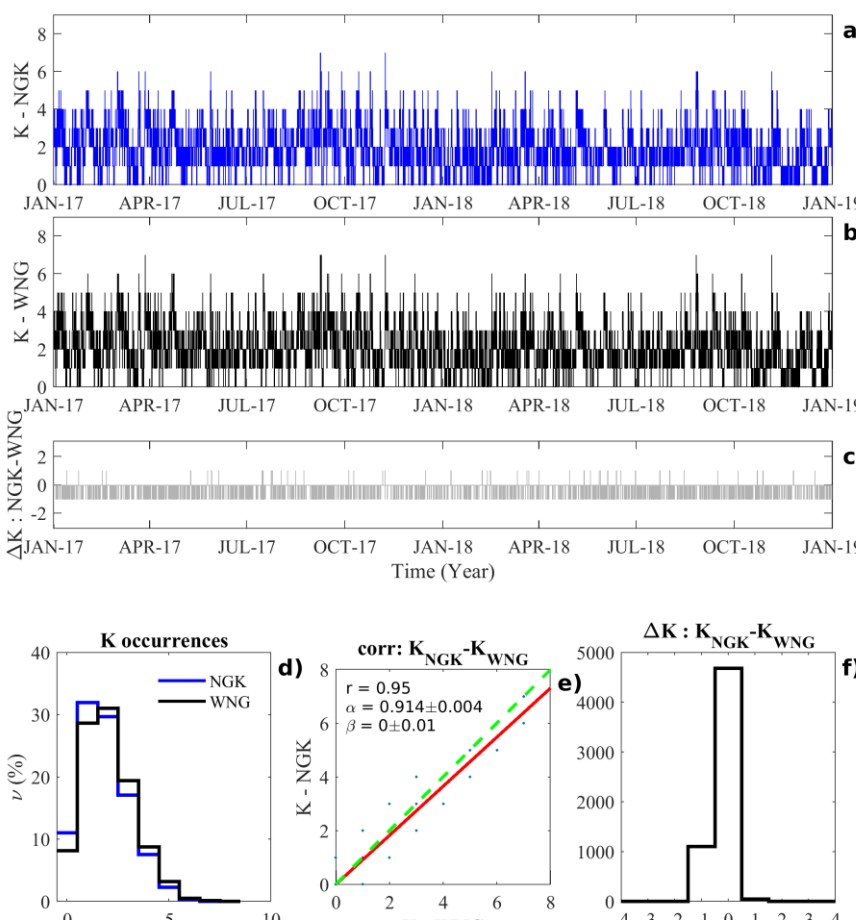

**Figure 2.** (top panels) $K$ indices at NGK (panel a) and WNG (panel b) and $\Delta K = K_{NGK} - K_{WNG}$ (panel c). Occurrences of $K$ indices at both observatories (panel d). Linear regression analysis of $K_{NGK} = \alpha K_{WNG} + \beta$ law (red line), by assuming $\beta = 0$, and correlation coefficient $r$, together with $K_{NGK} = K_{WNG}$ condition (green line). Panel f shows the $\Delta K$ distribution.

345





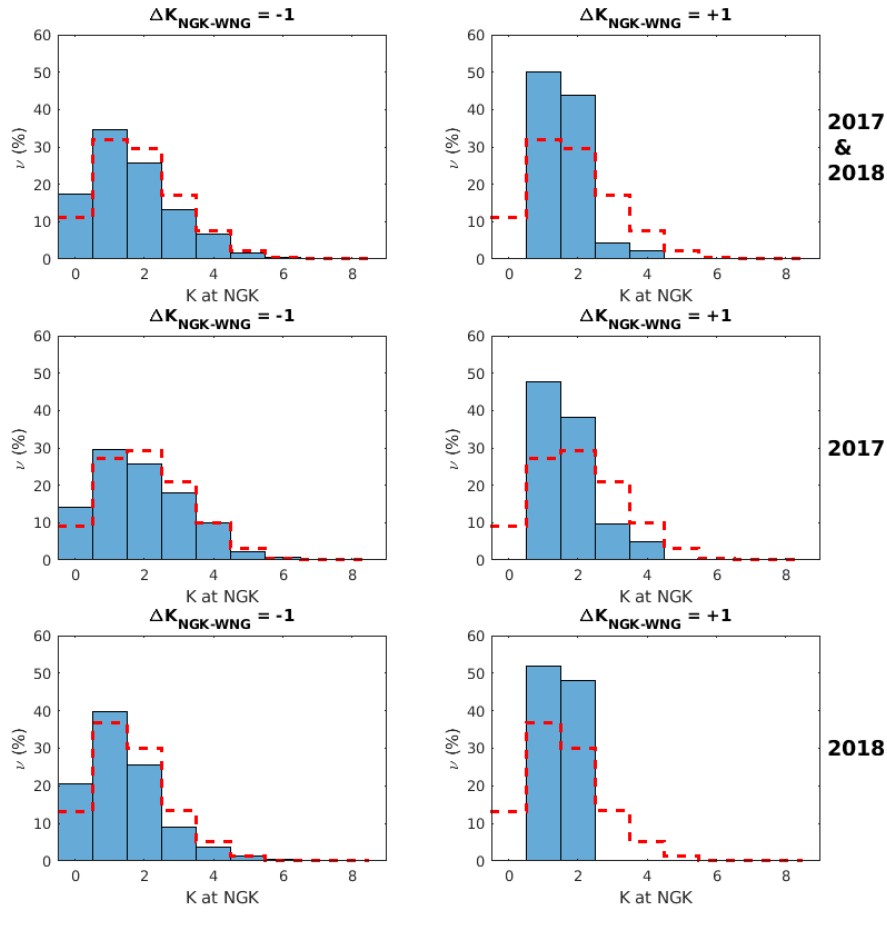

585

**Figure 3.** Frequency distributions of K index at NGK (red dashed line) superimposed on both distributions of $K_{NGK}$-$K_{WNG}$ = -1 (left side) and $K_{NGK}$-$K_{WNG}$ = 1 (right side) for analyzed years together (top panels) and separately for each year (middle and bottom panels).



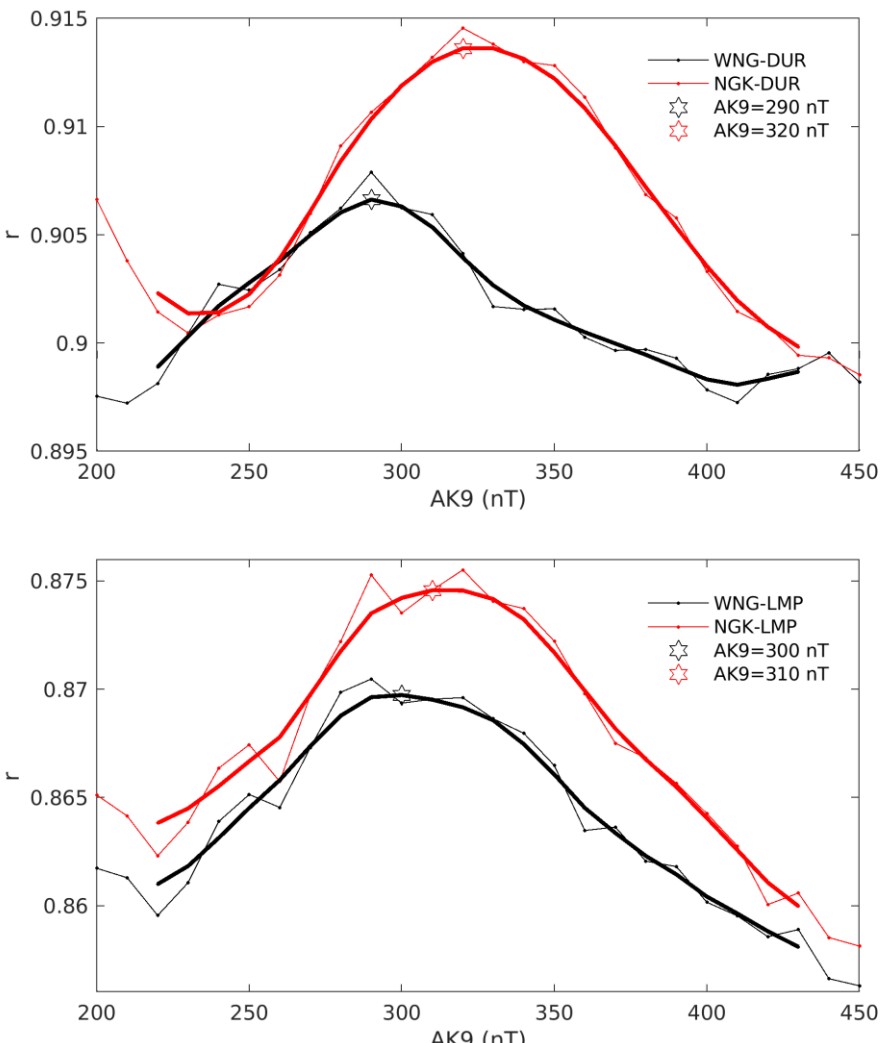

**Figure 4.** Correlation analyses between K index at NGK and that computed at DUR (top panel) and LMP (bottom panel) by KASM for different K9 values for the 2017-2018 dataset. The thick lines are obtained by smoothing the experimental (thin) lines. In each panel the maximum correlations, referring to the thick lines, are marked by stars (see text for details).





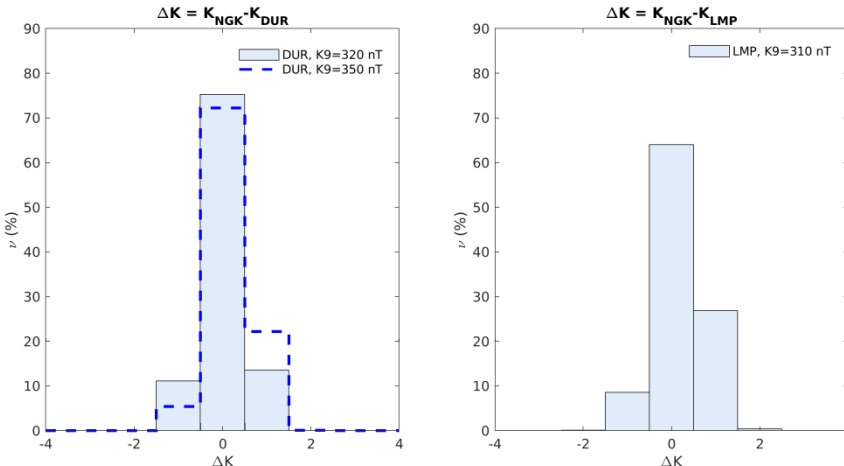

**Figure 5.** (left) Frequency distributions of $\Delta K = K_{NGK} - K_{DUR}$ for K9=320 nT and, for a comparison, the frequency distribution of $\Delta K = K_{NGK} - K_{DUR}$ for K9=350 nT. (right) $\Delta K = K_{NGK} - K_{LMP}$ (K9=310 nT).

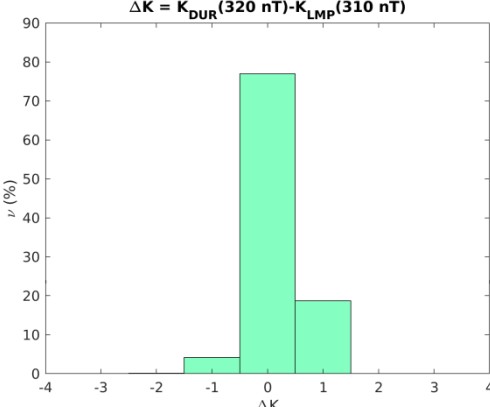

**Figure 6.** Frequency distribution of $\Delta K = K_{DUR} - K_{LMP}$, where K indices are computed by KASM by using K9=320 nT and K9=310 nT for DUR and LMP, respectively.

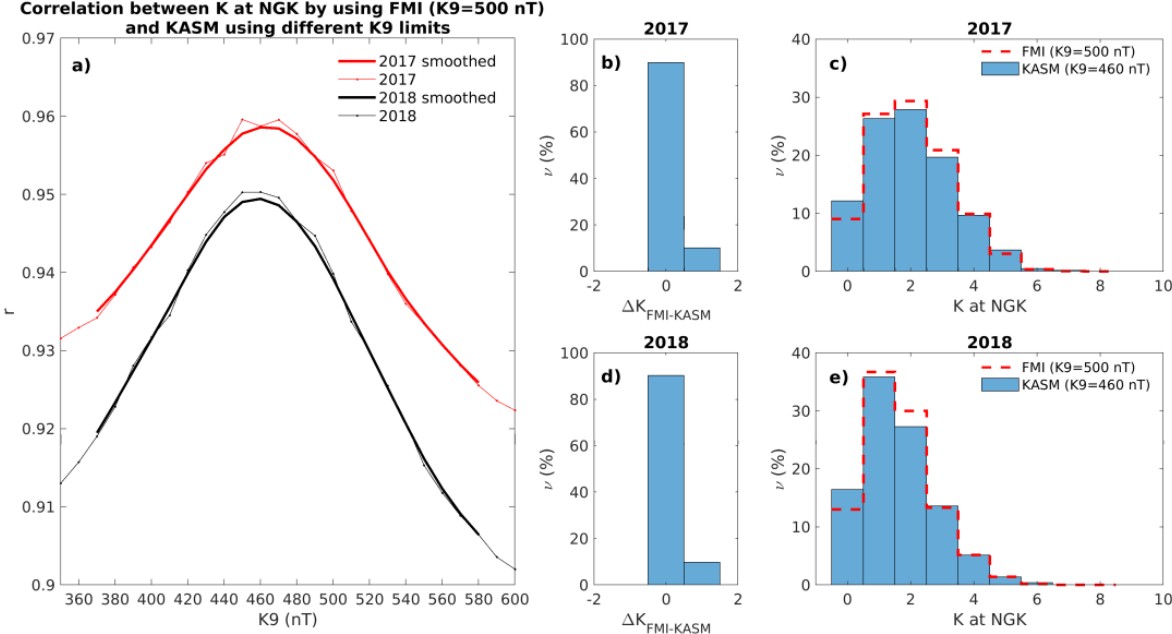

**Figure 7.** The FMI and KASM consistency test. (panel a) Correlation analyses between K index at NGK from FMI with K9=500 nT, and that from KASM for different K9 levels, for the years 2017 (definitive NGK 1 min data, red) and 2018 (quasi-definitive NGK 1 min data, black), respectively. The thick lines are obtained by smoothing the experimental (thin) lines. In each year the maximum correlations if found at K9=460 nT. The occurrences of $\Delta K=K_{FMI}-K_{KASM}$ during 2017 (b) and 2018 (d). Frequency distributions of $K_{KASM}$ (460 nT) and $K_{FMI}$ (500 nT) indices for the years 2017 (c) and 2018 (e).





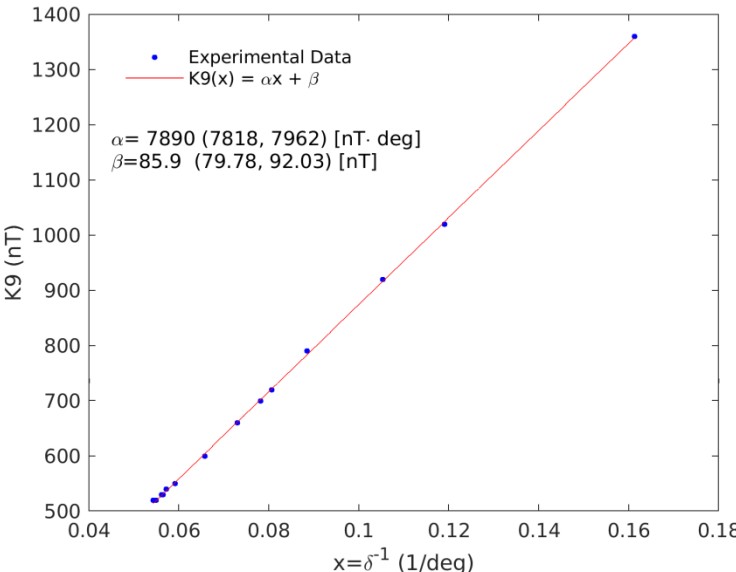

450

**Figure 8.** The linear relation between K9 limit and $x=1/\delta$ (Eq. 1), where the values are from Mayaud (1980). The linear regression fit results are shown.





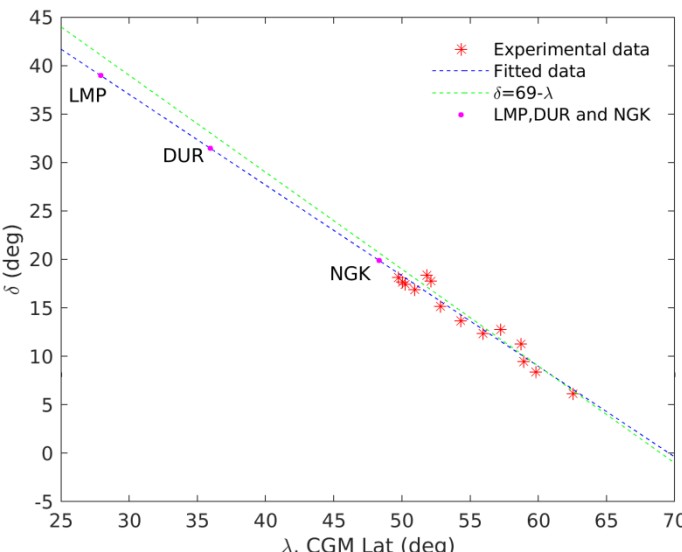

465    **Figure 9.** The linear relation between $\delta$ and the geomagnetic latitude $\lambda$ (Eq. 2), where the values from Mayaud (1980) are indicated by red
stars and the extrapolated values for our observatories by pink dots. The linear regression fit is shown in blue and the approximate relation
$\delta=69°\text{-}\lambda$ is shown in green.



470 **Table 1.** Geomagnetic observatories used in this study; geographic coordinates; Altitude Adjusted Corrected Geomagnetic Coordinates (AACGM), estimated at 100 km above the observatory and magnetic local time at 0 UT.

| Name | CODE | Geographic coordinates | AACGM coordinates | MLT (0 UT) |
|------|------|------------------------|-------------------|------------|
| Lampedusa | LMP | 35.52° N, 12.55° E | 27.9° N, 86.0° N | 00:29 |
| Duronia | DUR | 41.65° N, 14.47° E | 35.9° N, 88.5° E | 00:39 |
| Niemegk | NGK | 52.07° N, 12.68° E | 48.3° N, 88.9° E | 00:40 |
| Wingst | WNG | 53.74° N, 9.07° E | 50.2° N, 86.2° E | 00:30 |

475

**Table 2.** K9 estimated by different procedures. $\delta_a$=69°-$\lambda$ (approximate angular distance from auroral region); K9($\delta_a$) obtained from $\delta_a$ using linear fit in Fig.8; $\delta_{th}$ ($\delta$ estimated from AACGM lat and the linear fit in Fig. 9); K9($\delta_{th}$) obtained from $\delta_{th}$ using linear fit in Fig.8; K9$_A$($\Lambda$) obtained from correction procedure on the key parameter $\delta(\lambda)$ explained at the end of section 3.2. The 95% confidence intervals are also indicated.

| Name | $\delta_a$ [deg] | K9($\delta_a$) [nT] (Conf. Interv.) | $\delta_{th}$ [deg] | K9($\delta_{th}$) [nT] (Conf. Interv.) | K9$_A$($\Lambda$) [nT] (Conf. Interv.) | K9$_{exp}$ [nT] |
|------|------|------|------|------|------|------|
| LMP | 41 | 278 (269, 287) | 39 | 288 (268, 313) | 270 (248, 294) | 310 |
| DUR | 33 | 325 (315, 334) | 31 | 337 (312, 367) | 320 (295, 350) | 320 |
| NGK | 21 | 467 (459, 476) | 20 | 482 (441, 535) | 475 (432, 529) | 460 |

480