# Peer review of "On the validation of K index values at Italian geomagnetic observatories"

_Geoscientific Instrumentation, Methods and Data Systems, 2019_

## Referee Comment (RC1) · Anonymous Referee #1 · 18 Nov 2019

GENERAL COMMENTS

The paper addresses the problem of determining the K9-limit, which is the main input parameter for calculation of geomagnetic K indices. This problem is particularly important for newly established geomagnetic observatories in mid-latitudes, since K indices apply to such observatories. This is the case of Duronia and Lampedusa observatories the authors are connected with.

The broadly understood quality of K indices determined by a given observatory depends (among other things) on a correct determination of the K9-limit. The paper stresses correctly that this is a fundamental issue connected with K indices. Determining the appriopriate K9-limit at the beginning will result in correct K indices in the future.

The method proposed by the authors involves a statistical comparison of K indices of the new observatory with K indices from reference observatories. As reference observatories Niemegk and Wingst were selected. These observatories were selected since they belong to the observatories that contribute to the official Kp network, and their local magnetic time is close to Duronia and Lampedusa. The K9-limit for the new observatory was changed in a way ensuring maximal statistical correlation with reference observatories.

The experience on determining the K9-limit gained by the authors can be re-used by other observatories in the world. However, a problem may be to find good reference observatories, which fulfill at the same time two conditions. This is belonging to the Kp-netwok and their local magnetic time is close to the given observatory.

Another interesting undertaking of this work was to compare two computer methods of determining K indices, the ASM and FMI method. These are the methods most frequently used by geomagnetic observatories. The statistical studies show that both methods give similar results for slightly different K9-limits.

In general it can be summarized that the article is worth publishing. I believe that this paper can be interesting for many scientists in the world engaged in the work of geomagnetic observatories.

SPECIFIC COMMENTS

1)

In this paper is no information whether there was an attempt to obtain K9-limits for Duronia and Lampedusa from ISGI (International Service of Geomagnetic Indices). According to information on http://isgi.unistra.fr/isgi_refservice.php ISGI " has the responsibility of IAGA geomagnetic indices derivation and dissemination, and to ensure the homogeneity of the data series"

2)

In the paper "K9" or "K9 value" should be rather replaced with "K9-limit". The name "K9-limit" is closer to the idea of this parameter.

TECHNICAL CORRECTIONS

Line 45: on https://www.gfz-potsdam.de/en/kp-index/ we can see Lat=52deg4min (not 47.94deg)

Line 55: should be Finnish Meteorological Institute (not Meteorolical)

Line 55: should be LRNS provided by Hermanus Magnetic Observatory, CISR, South Africa

Line 280: consider deleting "the" or write "that they"

Line 289: should be 12b(89)

Line 303: Should be rather Geophysical Journal International

Line 312: Should be 10.1029/2018GL078387

---

## Referee Comment (RC2) · Anonymous Referee #2 · 10 Jan 2020

"On the validation of K index values at Italian geomagnetic observatories" by Mauro Regi et al.

The paper addresses a statistical method of determining the K=9 lower limit (L9) of magnetic observatories.
The results agree with the results given by the well-known method endorsed by IAGA and implemented by ISGI.
Nevertheless, some major corrections need to be made as well as a complete review of some parts of the bibliography.

*Main comments*

**M1-** ➔ Please, in the whole paper, state clearly how the geomagnetic / corrected geomagnetic / altitude adjusted corrected geomagnetic coordinates (latitude here) are determined.
- What is the software used?
- Is there a citation (DOI) of this software (e.g.: aacgm-v2 from Shepherd, S. G. (2014), Altitude-adjusted corrected geomagnetic coordinates: Definition and functional approximations, J. Geophys. Res., 119, 9, doi:10.1002/2014JA020264.])?
- Which underneath main field model is used (e.g; IGRF12? IGRF13?) ?
- What is the date used? (as any geomagnetic coordinates vary with time)

**M2-** "K=9 lower limit" is traditionally named "L9 value" or abbreviated as "L9". Indeed, it is the lower limit of classe K=9 at a particular observatory

*section Abstract :*

**A1-** *"The method for determining the K values should be the same for all observatories (…)"*
     ➔ Please, replace by "The method for determining the K values IS the same for all observatories."

**A2-** *"INTERMAGNET consortium recommends a software code, KASM (…)"*
This statement is incorrect. INTERMAGNET does not recommend KASM method.

     ➔ Please, correct according to the following: "INTERMAGNET recommend the use of one of the 4 methods recommended by ISGI (the International Service of Geomagnetic Indices) in close cooperation and agreement with the ad-hoc working group of International Association of Geomagnetism and Aeronomy."

The original definition of K indices (Bartels et al., 1939) requires hand scaling on analogue magnetograms. The question of the derivation of geomagnetic indices from digital data arose at the end of the seventies! Different algorithms enabling computer derivation of K indices were then developed and carefully assessed in the frame of an international comparison organised by the IAGA Working Group "Geomagnetic indices" (Coles & Menvielle, 1991; Menvielle, 1991; Menvielle et al., 1995).
See references:
- Coles, R., and M. Menvielle (1991) - Some thoughts concerning new digital magnetic indices, Geophys. Trans., 36, 303-312.
- Menvielle, M. (1991) - Evaluation of algorithms for computer production of K indices, Geophys. Trans., 36, 313 -320.

See your paragraph lines 46 to 58 in section 1-Introduction.

*section 1- "Introduction "*

**B1 - line 32 to 35:** *"Therefore, K index is the fundamental parameter for Kp estimation that is widely used, for example, in space weather applications, for identify quietest days (Johnston, 1943) used also in the IGRF modeling, for verifying solar wind driven modulation in the atmospheric parameters during disturbed conditions (Regi et al., 2017)."*
This paragraph appears only as a way to allow citation of (Regi et al., 2017).
Kp is a K-derived geomagnetic index at sub-auroral latitudes only. Furthermore, even if Kp purpose was to characterize the intensity of geomagnetic activity on a planetary scale, authors have to be pragmatic. Kp was developed in other times and, because of the historical context at the time of its creation (cold war), the Kp network is heavily weighted towards Europe and Northern America.
The citation of a paper presenting a study « ULF geomagnetic activity effects on tropospheric temperature, specific humidity, and cloud cover in Antarctica, during 2003–2010 » is not a proper example here. Antarctica being far away from sub-auroral and Northern hemisphere.
     ➔ Please, explain clearly the fact that Kp is an historical index with known drawbacks and erase the citation to Regi et al.

**B2 - line 42 to 43**: *"For example (see Bartels et al., 1939), from higher to lower latitude, at Sitka (AACGM latitude λ=52° N, Alaska), K9=1000 nT, while in Honolulu (λ=21.37° N, Hawaii) K9=300 nT. The GFZ website (https://www.gfz-potsdam.de/en/kp-index/) provides K9 values for the 13 observatories used for Kp evaluation, showing values between 450 nT and 1500 nT; in particular, at Niemegk (λ=47.94° N, Germany) K9=500 nT."*

➔ Please, see M1regarding description of coordinates.

L9 used are different from the L9 determined/calculated.
That fact came from history. In the middle of 20th century, the aim was on one hand, to avoid to constrain the observers of magnetic observatories (to acknowledge their skills and free will), and on the other hand, to let a possible rounding of L9. At that time, when calculations were done by hand and K indices were hand-scaled, differences of some tenths of nT were not a big deal.

Example of Lerwick:
     L9 used      = 1 000 nT for Kp but
     L9 calculated = 921 nT

Indeed, the observers of each observatory were set free to "round" the values :
- towards the "nearest" decade of nT (921 to 920),
- towards the "nearest" fifty of nT (921 to 950 or 900)
- towards the "nearest" hundred of nT (921 to 900 or 1000)

The L9 presented onto the GFZ website are the L9 used for the calculation of Kp.

➔ Please, clearly state here when the L9 are the ones used for historical purposes (derived at the end of the forties by Bartels et al., only with hand-scaling) or the ones calculated and determined by ISGI under the auspices of IAGA, in agreement with the international community in geomagnetism.
The L9 you are showing are mainly the ones used for the Kp data series, to remain consistent along time, Honolulu being not used in Kp calculation but only to show a low latitude example.
➔Please, correct the sentences, for example: "For Kp determination (Bartels et al., 1939), from higher to lower latitude, at Sitka ([coordinates given], Alaska), L9 =1000 nT while Canberra ([coordinates given] , Alaska), L9 =500 nT. The GFZ website (https://www.gfz-potsdam.de/en/kp-index/) provides the 13 L9 values used for the Kp evaluation, showing values between 450 nT and 1500 nT; in particular, at Niemegk ([coordinates given], Germany) L9=500 nT."

**B3 - Lines 46 to 58 :** *"For many years, K was manually derived by means of a conversion table containing the values of the maximum fluctuation A, expressed in units of nT, for each K value. With the introduction of digitized data and with the increasing access to computers, the manual estimation of K index was progressively substituted with automated algorithms and, nowadays, the reproducibility, one of the cornerstones of science, has become possible. (…) http://isgi.unistra.fr/softwares.php)."*

➔ Please, correct or amend the first sentence: "For many years, K was manually scaled by means of visual determination of the regular daily variation and of the consequent largest range of geomagnetic disturbances in the two horizontal components during a 3-hour UT interval. Then, K indices were determined by means of a conversion table between classes of ranges in nT and K indices."

➔ Please, enclose and introduce the two following missing, but fundamental, references:

- Coles, R., and M. Menvielle (1991) - Some thoughts concerning new digital magnetic indices, Geophys. Trans., 36, 303-312.
- Menvielle, M. (1991) - Evaluation of algorithms for computer production of K indices, Geophys. Trans., 36, 313 -320.

**B4 - Lines 59 to 61:** *"The International Real-time Magnetic Observatory Network (INTERMAGNET, http://www.intermagnet.org), of which IAGA is associated, endorses and recommends KASM for calculation of geomagnetic activity indices K according to the Adaptive Smoothed method (Nowozyński et al., 1991)."*

➔ Please, see comment A2. This sentence is incorrect, it has to be replaced by: "IAGA, through the ISGI international service, endorsed 4 different methods for calculation of local geomagnetic activity indices K. We used one of them the KASM method that used adaptive smoothed method (Nowozyński et al., 1991)."

**B5 - Line 63:** "(…) the code derives daily values without fluctuations (mainly daily variation)."

➔ Please correct the wording. "(…) the code estimates the regular daily variation."

**B6 - Lines 65 to 70:** *"We want to point out that it does not exist an unique K9 at a given geomagnetic latitude since the geomagnetic activity shows a well known magnetic local time (MLT) dependency and, in addition, each site could be affected by different local features such as, for example, crustal anomalies (Chiappini et al., 2000) and/or coast effect (Parkinson, 1962; Regi et al., 2018). For the inclusion of a new geomagnetic observatory into the INTERMAGNET network, K9 should be assigned, for example, by comparing geomagnetic field variations between the new observatory and the historical ones for which K indices are estimated by using well defined K9 levels, obtained from a long time observation."*

➔ **This paragraph is entirely false.** It does exist a unique L9 at a given geomagnetic latitude. The 4 softwares endorsed by IAGA are taking care of the determination of the regular daily variations and are, by construction, considering the day-to-day variability. A simple plot of the regular daily variations extracted from softwares shows it clearly. Although one has to dig into the code and extract the relevant information. Indeed, codes available at ISGI are designed for operational purposes and were designed considering that the user knows their internal functioning.

At a particular magnetic observatory L9 is defined by the distance $\delta$ to "oval auroral" modelled as the +/-69° latitude CGM around 1965.

See:

- Mayaud, P.-N. (1968) - Indices Kn, Ks et Km, 1964-1967, Ed. C.N.R.S., Paris, 156 p.

Especially the Figure A1 page 34.

A more recent paper is using that historical reference and may appear less 'arid' to the authors of the present paper:

- Lockwood, M., A. Chambodut, L. A. Barnard, M. J. Owens, E. Clarke, and V. Mendel. 'A Homogeneous Aa Index: 1. Secular Variation'. Journal of Space Weather and Space Climate 8 (2018): A53, doi: 10.1051/swsc/2018038.

Especially the Figure 3 page 6 and the related section 2.

Bartels defined the L9 of a particular magnetic observatory from hand-scaling of ranges and subsequent statistical study with the intent of producing a geomagnetic disturbance characterisation that does not depend significantly on the location of a sub-auroral, mid- or low- latitude observatory.

However, since beginning of the fifties, and even more since the digital era, the empirical method was put apart and the definition of L9 was chosen with regard to distance $\delta$ to "oval auroral" modelled as the +/-69° latitude CGM. (One has to note that this method remained still unperfect, as the distance to the oval auroral is changing with the main field but this is beyond the scope of your paper here.)

**B7 – line 79**: *" LMP is the southernmost observatory in Europe"*

➔ Please, correct this statement which is false GUI (Guimar-Tenerife) magnetic observatory is southernmost, not speaking about French austral territories…

**B8 – lines 88 to 90:** *"Our investigations suggest that NGK is the best reference observatory for Italian geomagnetic observatory of DUR, probably due to the closest magnetic local times: by comparing DUR with NGK we estimated a reliable DUR K9 level of 320 nT. Finally, by comparing also LMP with NGK, a reliable LMP K9 level of 310 nT is estimated."*

A simple computing, considering distance to the oval auroral, leads to:

- for historical determination (without secular variation, Mayaud's method) L9 $_{DUR}$ = 356 nT and L9 $_{LMP}$ = 315 nT;
- for January 2019 determination (taking the oval auroral given by IGRF) L9 $_{DUR}$ = 354 nT and L9 $_{LMP}$ = 312 nT.

The results agree with the one provided by the authors. (The agreement is less striking for DUR as its geomagnetic latitude is beyond the range of possible K indices determination.) CGM coordinates of the Italian magnetic observatories remain quite constant along time as the L9$_{NGK}$ does.

➔ Please, correct or explain the part of the sentence saying that "*probably due to the closest magnetic local times*". If the L9 values are wisely chosen, then, local K indices statistical repartition along K values does not depend significantly on the location of the observatory. But in any case, a comparison along time obviously does show a clear Local Time (or Magnetic Local Time) dependence as the magnetic disturbances are impacting differently the day, dawn, dusk or midnight quarter (e.g. : K-derived magnetic indices in 4 Magnetic Local Time sectors ; see Chambodut, A., A. Marchaudon, M. Menvielle, F. El-Lemdani and C. Lathuillere (2013) - The K-derived MLT sector geomagnetic indices, Geophys. Res. Lett., 40, 4808-4812, DOI:10.1002/grl.50947.)

**section 2-"Data and methods of analysis"**

**C1-** Please, which time-resolution magnetic observatory data are you using with the KASM method? Please indicate it. A first guess would be "minute data computed from second data using INTERMAGNET 1s to 1min filter".

**section 3-"Experimental results"**

**D1- regarding Figure 3 and the related description in the section.** The discrepancies observed are for low K indices. Do the authors have an explanation? Can it be a limitation of the K index derivation scheme in really quiet and quiet magnetic conditions?.

**D2- regarding Figure 4 and the related description in the section 2.** The discrepancies observed between the black and red curves are, for both compared observatories (LMP and DUR), in the same direction. WNG comparison shows an underestimation of L9 values. Would it be possible that the location of WNG observatory nearby the shore (around 10 km to the North sea) leads to a possible bias in daily regular variation estimation in K indices calculation?

**D3- lines 146 to 150:** *"In addition, the higher correlations are obtained by using NGK, probably due to the lower latitude (i.e. closer to the Italian observatories) and the closer MLT with respect to DUR (table 1). Also at LMP, even if the MLT is closest to that of WNG, the higher correlation is found with NGK: this result suggests that latitudinal effects are dominant with respect to MLT ones. This can be well understood taking into account that the MLT range of all selected observatories is within 11 minutes, well shorter than the 3-hour interval used for K determination."*
→ Please, consider D1 and D2 questions.

**D4-lines 179 to 180:** *"We point out how the distributions are close to each other, suggesting that FMI and KASM are consistent algorithms, (…)"*
You obtained the same results as: Coles & Menvielle (1991), Menvielle (1991) and Menvielle et al. (1995).

**D5- regarding Figure 5 and the related description in the section 2**:
→Please explain why Figure 5 (left) has to be symmetric for a better L9 value considering that only 2 years of data are used, a tiny part of the solar cycle?

**D6- regarding Figure 7 and the related description in section 2:** The authors are in fact here doing a comparison of K derivation softwares. The discrepancies observed are similar with the ones observed between Asm method and FMI method in Menvielle et al. (1995).

*subsection 3.2-" Comparison with a previous K9 estimation method"*

**E1- lines 190 and 191:** *"According to Mayaud (1980), an approximate value of δ could be given by δ=69°-λ but this is really just a rough approximation."*
→ Please, do not be so rude. Mayaud method is still the one in use that proved, and still proves, its robustness. The results of the present paper are similar to the ones obtained with Mayaud's method, see B8.
Furthermore, the correct reference here is Mayaud, P.-N. (1968) - Indices Kn, Ks et Km, 1964-1967, Ed. C.N.R.S., Paris, 156p. (available at http://isgi.unistra.fr/Documents/Books/Mayaud_CNRS_1968_complete.pdf)

**E2-lines 193 to 200:**
→ Please correct the approximated equation you are using. Mayaud is using a 4[th] degree polynomial.

**E3 lines 192 to 231:**
→ This part of the present paper is largely incorrect. Please, read Mayaud (1968) (or Lockwood et al. (2018), page 5 to 7, for a more "modern english" explanation) and correct.

**F1-lines 234 to 236:** *"The modern automatic procedures for calculating local K index values, with the setting of some a-priori criteria, have to be carefully verified for their permanent validation in terms of accuracy and stability when delivered to the scientific community."*
→ Please erase this sentence. The present paper has to be reviewed with major revisions.

**F2- line 237**: *"(…) This code is distributed by the INTERMAGNET consortium(…)"*
→ Please see comments A2 and B4.

**F3- lines 239 to 241:** *"(…) the K9 value, which represents the minimum value of the amplitude extent in the H component of Earth's magnetic field when the local K value reaches the integer 9, the highest level in a scale which ranges from 0 to 9."*
→ Please correct: "L9 value, the so-called "K=9 lower limit" allows to determine, for each magnetic observatory, the conversion table between classes of ranges and K indices."

**F4- lines 263 to 272:** *"Moreover, Mayaud (1980) note that the limitation of the method they propose is that it is conceived for sub-auroral and mid latitudes; indeed, they suggest that for lower latitudes a constant K9=300 nT can be chosen. This very approximate value is not very far from the values we estimate (320 nT for DUR and 310 nT for LMP), but would certainly be not accurate as them in the comparison with the values from other reference observatories: indeed our results clearly show that a very precise K9 limit is necessary for obtaining K values well consistent at different sites. As a final remark, from the overall view of this work, we are also definitely convinced that the habit to round the value of K9 in multiples of 50 nT is a simplified approximation, firstly suggested by Bartels et al. (1939), a practice that needs to be abandoned. This approximation is still adopted in some cases, demonstrating that perhaps a critical revision has not been applied yet, differently from the case of Kakioka observatory (Japan) where K9 has a convincing value of 296 nT."*

➔ Please erase that part. Mayaud (1968) clearly stated that the magnetic observatories towards polar areas or towards equatorial regions ( 58° > |CGM latitude| > 29°) are under magnetic conditions (e.g.: field aligned currents, magnetospheric ring current, etc) that do not allow to produce K indices comparable to mid-latitude ones. The Figure A1 of Mayaud (1968) clearly show a hyperbola with two asymptotes. One may calculate K indices for sub-equatorial or sub-polar geomagnetic observatories but without real physical meaning. The activity of the magnetic field in these observatories cannot be assessed with the proposed softwares. This is also the reason why K-derived Magnetic indices, such as Kp, aa or am, are only fully meaningful for mid-latitude. Dst (equatorial) or PC (polar) are not K-derived indices.

➔ Please, do not patronise and give recommendation. Indeed, the L9 values are presented onto the ISGI website for each magnetic observatory. The rounding of L9 values may easily be overcome by the use of the well-know FKC table developed and used by both Bartels and Mayaud. Magnetic Observatories do not need to update or change their L9 values. Homogeneity of the series is of primary importance.

For each magnetic observatory, the only mandatory point and message that should be given to the whole community is: Please, provide carefully in the metadata the L9 value that was actually used for K indices calculation.

---

## Referee Comment (RC3) · Anonymous Referee #3 · 3 Feb 2020

I evaluate favorably this paper because there is a serious work to assess the convenience of maintaining the k9 values given by IAGA to the Italian observatories. It has interesting discussions about the consequences of using different algorithms to compute local k indices.

I find that the paper is suitable for publication subject to some edits. I have some specific comments for the authors but also have several questions that the authors should address in a revised manuscript.

Firstly, there are two major questions where, from my point of view, the authors are out of focus and should be realigned in this paper.

a) The paper presents the situation as if each observatory has to choose its own k9.

[Figure]

But, according to the International Service of Geomagnetic Indices (ISGI), the ISGI-headquarters are in charge of the computation of k9 value for each magnetic observatory. ISG is a service of the International Association of Geomagnetism and Aeronomy (IAGA) which recognized "the unique role of the International Service of Geomagnetic Indices (ISGI) in the derivation, publication, and dissemination of these indices" (IAGA, Resolution No. 9 (1989). So, this paper should present its conclusions as an academic effort to check the assigned k9 values rather than to propose a new operational value.

b) K index is a coarse indicator of magnetic activity which simplifies the environmental situation into 10 digits for the sake of having a simple way to classify a magnetic disturbance. The scale is not linear nor logarithmic but a sort of personal choice of Bartels to have some events in each interval. Specially, in the high numbers, very diverse disturbances are assigned to the same figure (e.g. k=8 in Nimeck could be 331 nT or 500 nT) Moreover, Sq estimation, necessary because it should be removed before k computation, is rather subjective and involve a large uncertainty. In each algorithm, Sq is interpreted differently (Menvielle et al., 1995).

The authors claim the necessity of giving k9 values in units of nT or even with tenths of nT (Ln 268-273) but this it would be misinterpreted as if they were very precise when they are not!

On the other hand, other minor questions are addressed to the authors:

c) Niemek (NGK) was the reference observatory where this scale was defined by Bartels. The rest of other observatories where assimilate to this to create distributions similar to that one. So, the comparison of the Italian observatories with this observatory has more sense than with other German observatory (WNG). In fact, this observatory although being located very close to NGK, has not a perfect correspondence in distribution with (19% deltaK=-1, ln 130).

d) However, as K index measures the effect of auroral activity, it seems more reasonable to compare Italian observatories' distribution with other observatories with similar

latitude. Moreover, it is well known that k9 limit does not follow a regular law with the angular distance to the auroral zone (Mayaud, 1980).

e) Although it is true that digital algorithms grant reproducibility (Pg 2 ln 49), this does not mean that they are more certain. In the past, K index was "estimated" for manual procedures; but, now, an automatic algorithm also produces "estimated" values. And, of course, different algorithms would produce different values (pg9 ln255).

f) Comparing the distribution of Italian indices generated by KASM algorithm with NGK and WNG indices generated by FMI algorithm (Ln 137) is a rough way to do this because these distributions change with the algorithm and even with the year being considered (Figure 7). h) The correlation analysis used to obtain the best value of k9 (figure 4) presents a flat and asymmetric shape in an interval ranking for more than 50 nT. There, any change, as the use of a different algorithm, would produce a different maximum. So, I would not take the new values as a step forward. In fact, final results (those choosing a new value of k9 for DUR (fig. 5) implies a variation of 10% of population in deltaK=-/+1, in the limit of the precision of the method.

---

## Author Response (AR1)

**Replies to Reviewer #1**

The Authors thank the reviewer for her/his helpful comments which increased the quality of our work. Here below we reported the reviewer's comments (in black) and our replies (in red).

**SPECIFIC COMMENTS**

1) In this paper is no information whether there was an attempt to obtain K9-limits for Duronia and Lampedusa from ISGI (International Service of Geomagnetic Indices). According to information on http://isgi.unistra.fr/isgi\_refservice.php ISGI " has the responsibility of IAGA geomagnetic indices derivation and dissemination, and to ensure the homogeneity of the data series".

In the first version of the paper the authors had a private communication from colleagues who are connected with ISGI regarding the K9 value for Lampedusa observatory. In the revision process we indirectly obtained the K9 value for Duronia observatory. Both values are discussed in the revised version of the paper for corroborating our results (lines 265-267 in the revised manuscript).

2) In the paper "K9" or "K9 value" should be rather replaced with "K9-limit". The name "K9-limit" is closer to the idea of this parameter.

According to the #2 referee's suggestions the authors made use of L9 (this is how the parameter is traditionally named) instead of K9 in the revised manuscript.

**TECHNICAL CORRECTIONS**

Line 45: on https://www.gfz-potsdam.de/en/kp-index/ we can see Lat=52deg4min (not 47.94deg)

We corrected the sentences also following the #2 reviewer's indications (lines 50-53, revised ms).

Line 55: should be Finnish Meteorological Institute (not Meteorolical)

Done: line 64 in the revised manuscript

Line 55: should be LRNS provided by Hermanus Magnetic Observatory, CISR, South Africa

Done: line 64 in the revised manuscript

Line 280: consider deleting "the" or write "that they"

Done: we used "that they" at line 300 in the revised manuscript

Line 289: should be 12b(89)

Done.

Line 303: Should be rather Geophysical Journal International

Done

Line 312: Should be 10.1029/2018GL078387

Done

**Reply to reviewer#2**

The Authors thank the reviewer for her/his helpful comments which increased the quality of our work. Here below we reported the reviewer's comments (in black) and our replies (in red).

The paper addresses a statistical method of determining the K=9 lower limit (L9) of magnetic observatories. The results agree with the results given by the well-known method endorsed by IAGA and implemented by ISGI. Nevertheless, some major corrections need to be made as well as a complete review of some parts of the bibliography.

**Main comments**

**M1-** Please, in the whole paper, state clearly how the geomagnetic / corrected geomagnetic / altitude adjusted corrected geomagnetic coordinates (latitude here) are determined.

- What is the software used?
- Is there a citation (DOI) of this software (e.g.: aacgm-v2 from Shepherd, S. G. (2014), Altitude-adjusted corrected geomagnetic coordinates: Definition and functional approximations, J. Geophys. Res., 119, 9, 7501–7521, doi:10.1002/2014JA020264.])?
- Yes, we used the aacgm-v2 algorithm. We properly include the reference in the revised version of the manuscript (line 50 of the revised ms).
- Which underneath main field model is used (e.g; IGRF12? IGRF13?)?
  - We never directly used the IGRF model. We generally mention the IGRF at lines 37 and 218 (revised ms) regardless the IGRF version .
- What is the date used? (as any geomagnetic coordinates vary with time)

We referred our computing to 2017 as specified in line 50 (see revised ms).

M2- "K=9 lower limit" is traditionally named "L9 value" or abbreviated as "L9". Indeed, it is the lower limit of classe K=9 at a particular observatory

Thank you, we have substituted "K9" with "L9" everywhere in the manuscript (main body of the manuscript, figures, tables and captions).

**section Abstract :**

A1-"The method for determining the K values should be the same for all observatories (...)"

→ Please, replace by "The method for determining the K values IS the same for all observatories."

The referee is right; we prefer to refine this sentence to: " The method for determining the K values HAS TO BE the same for all observatories."

**A2- "INTERMAGNET consortium recommends a software code, KASM (...)"**

This statement is incorrect. INTERMAGNET does not recommend KASM method.

→ Please, correct according to the following: "INTERMAGNET recommends the use of one of the 4 methods recommended by ISGI (the International Service of Geomagnetic Indices) in close cooperation and agreement with the ad-hoc working group of International Association of Geomagnetism and Aeronomy."

We modified the sentence according to the reviewer suggestion. However, we note what is published in the INTERMAGNET website at the "software" subpage (https://www.intermagnet.org/publication- software/software-eng.php, see below its screenshot). It clearly states that "INTERMAGNET does not endorse or recommend any of the non INTERMAGNET software" and the only software provided for computing K indices is KASM.

**INTERMAGNET**

TERMAGNET Data Observatories (IMOs) Participating Institutes Publications/Softwares How to Reach Us

| Home > Software            |                                                                            |  |  |  |  |
|----------------------------|----------------------------------------------------------------------------|--|--|--|--|
| Technical Reference Manual | Software                                                                   |  |  |  |  |
| Publications               |                                                                            |  |  |  |  |
| Meetings                   | u                                                                          |  |  |  |  |
| Software                   |                                                                            |  |  |  |  |
| Yearbooks                  | Content now located at https://intermagnet.github.lo/somware.html . |  |  |  |  |
|                            |                                                                            |  |  |  |  |

INTERMAGNET does not endorse or recommend any of the non-INTERMAGNET software. These applications are here to help the community find tools that might be useful to them.

**INTERMAGNET GitHub @**

A collection of source code including routines in Python, Mathematica, Matlab, IDL and Java for reading and writing the INTERMAGNET CDF format ImagCDF. This repository is open for other source code that would be useful to the community and we encourage contributions.

**MagPY 🖉**

MagPy (or GeomagPy) is a Python package for analysing and displaying geomagnetic data.

**Imcdview 🗗**

Imodview is an application for working with the CDs and DVDs of 1-minute definitive data that INTERMAGNET has published annually since 1991. The programme is compiled and will run on Microsoft Windows, SOLARIS, Linux and OS X (source code is not included). The program allows you to view minute, hourly and daily mean data and to scroll rapidly through data sets, and also to view quality control information such as comparison of instruments at an observatory, comparison of data between nearby observatories and plots of the 'baseline' at an observatory - an important indicator of the quality of measurements. Metdata such as observatory location and contact details for institutes is also available.

Imodview works with the INTERMAGNET Archive Format (IAF) and requires data to be in the structure defined for the INTERMAGNET CD/DVD. One module in the programme allows import of data from other formats (such as IAGA-2002) into the IAF format. Another module provides options to export the data in a number of geomagnetic data formats. Caution should be used when working with these import and export facilities - data will always be limited to the precision of the IAF format, which is particularly low for angles (declination).

**DataCheck1S**

DataCheck1s is primarily intended for INTERMAGNET members who are tasked with quality checking INTERMAGNET 1-second data. The programme compares 1-second data in IAGA-2002 format to 1-minute data from the same observatory / time range in IAF format. It also converts data from IAGA-2002 to the new INTERMAGNET CDF format (ImagCDF) and provides some simple plotting abilities.

**Autoplot 🖉**

Autoplot is a general purpose plotting package which can be used to view data in INTERMAGNET's CDF format (ImagCDF) as well as a number of other formats. It is able to handle large volumes of time series data efficiently and provides spectral as well as time series plots.

Autoplot was developed under the NASA Virtual Observatories for Heliophysics program in a collaborative effort among several institutions, including support or code contributions from ViRBO, VMO, RBSP-ECT, and the Radio and Plasma Wave Group at The University of Iowa.

**check1min 🖻**

Software to check INTERMAGNET 1 minute definitive data, as described in the INTERMAGNET technical manual (ver 5).

**GeomagLogger**

A C++ software package for logging data at geomagnetic observatories.

**MagPySV**

Python toolbox for geomagnetic data processing particularly related to SV work.

**Kasm**

Program Kasm is designed for calculation of geomagnetic activity indices K according to the Adaptative Smoothed method.

**Gm\_convert**

This software allows you to convert between a number of geomagnetic formats. The program can read the following formats: WDC; INTERMAGNET Minute Mean Format; IAGA-2002; INTERMAGENT Archive Format; INTERAMGNET-CDF; INTERMAGNET-DKA. It converts to either INTERMAGNET Archive Format; IAGA-2002 or INTERAMGNET-CDF. The program can convert from large numbers of input files in multiple formats. It is designed so that, once the necessary information has been collected from the user, the conversion will take place without requiring further input, meaning that long conversion tasks can run unattended.

Date modified: 2020-02-02

Contact Us

Webmaster (NRCan.geomag-webmaster-

The original definition of K indices (Bartels et al., 1939) requires hand scaling on analogue magnetograms. The question of the derivation of geomagnetic indices from digital data arose at the end of the seventies! Different algorithms enabling computer derivation of K indices were then developed and carefully assessed in the frame of an international comparison organised by the IAGA Working Group "Geomagnetic indices" (Coles & Menvielle, 1991; Menvielle, 1991; Menvielle et al., 1995).

See references:

- Coles, R., and M. Menvielle (1991) Some thoughts concerning new digital magnetic indices, Geophys. Trans., 36, 303-312.
- Menvielle, M. (1991) Evaluation of algorithms for computer production of K indices, Geophys. Trans., 36, 313 -320.

Thank you for suggesting how to introduce the reader to the passage from analogue to digital evaluation. We included this useful discussion in the lines 51-58 (revised ms), as well as the suggested references (lines 194-195 in the revised ms).

**section 1- "Introduction "**

**B1 - line 32 to 35:** "Therefore, K index is the fundamental parameter for Kp estimation that is widely used, for example, in space weather applications, for identify quietest days (Johnston, 1943) used also in the IGRF modeling, for verifying solar wind driven modulation in the atmospheric parameters during disturbed conditions (Regi et al., 2017)."

This paragraph appears only as a way to allow citation of (Regi et al., 2017). Kp is a K-derived geomagnetic index at sub-auroral latitudes only. Furthermore, even if Kp purpose was to characterize the intensity of geomagnetic activity on a planetary scale, authors have to be pragmatic. Kp was developed in other times and, because of the historical context at the time of its creation (cold war), the Kp network is heavily weighted towards Europe and Northern America. The citation of a paper presenting a study « ULF geomagnetic activity effects on tropospheric temperature, specific humidity, and cloud cover in Antarctica, during 2003–2010 » is not a proper example here. Antarctica being far away from sub-auroral and Northern hemisphere.

→ Please, explain clearly the fact that Kp is an historical index with known drawbacks and erase the citation to Regi et.

We aim to have a scientific, not pragmatic approach.

The referee in right in asserting that Kp has known drawbacks, and we added this consideration in the revised ms (lines 35-36). However Kp, even if computed at sub-auroral latitudes, is important to characterize the planetary geomagnetic activity, and consequently is widely used by scientific communities in relation to both magnetospheric and ionospheric/geomagnetic domains, regardless the geomagnetic latitude. Auroral activity indexes such as AE and AO are obviously better related with polar cap electrodynamics with some limitations: there are not long time series since they have been only recently introduced and they are not available in near real -time.

Regi et al. showed that tropospheric and stratospheric parameters are affected by sudden planetary geomagnetic activity changes, as geomagnetic storms occurred on October 2003 (Halloween superstorm) and 27 July 2004, using both Kp and AE indices to characterize the geomagnetic activity. Therefore we believe that citing this paper is an appropriate example for make evident the importance of K index in the Space-Weather and Space-Climate context.

**B2 - line 42 to 43:** "For example (see Bartels et al., 1939), from higher to lower latitude, at Sitka (AACGM latitude  $\lambda$ =52° N, Alaska), K9=1000 nT, while in Honolulu ( $\lambda$ =21.37° N, Hawaii) K9=300 nT. The GFZ website (https://www.gfz-potsdam.de/en/kp-index/) provides K9 values for the 13 observatories used for Kp evaluation, showing values between 450 nT and 1500 nT; in particular, at Niemegk ( $\lambda$ =47.94° N, Germany) K9=500 nT."

➔ Please, see M1 regarding description of coordinates.
Please see our reply to M1.

L9 used are different from the L9 determined/calculated.

That fact came from history. In the middle of 20th century, the aim was on one hand, to avoid to constrain the observers of magnetic observatories (to acknowledge their skills and free will), and on the other hand, to let a

possible rounding of L9. At that time, when calculations were done by hand and K indices were hand-scaled, differences of some tenths of nT were not a big deal.

Example of Lerwick:

L9 used = 1 000 nT for Kp but

L9 calculated = 921 nT

Indeed, the observers of each observatory were set free to "round" the values :

- towards the "nearest" decade of nT (921 to 920),

- towards the "nearest" fifty of nT (921 to 950 or 900)

- towards the "nearest" hundred of nT (921 to 900 or 1000)

The L9 presented onto the GFZ website are the L9 used for the calculation of Kp.

Please, clearly state here when the L9 are the ones used for historical purposes (derived at the end of the forties by Bartels et al., only with hand-scaling) or the ones calculated and determined by ISGI under the auspices of IAGA, in agreement with the international community in geomagnetism.

The L9 you are showing are mainly the ones used for the Kp data series, to remain consistent along time, Honolulu being not used in Kp calculation but only to show a low latitude example.

This is an interesting remark we added at lines 289-293 of the revised manuscript.

→ Please, correct the sentences, for example: "For Kp determination (Bartels et al., 1939), from higher to lower latitude, at Sitka ([coordinates given], Alaska), L9 =1000 nT while Canberra ([coordinates given], Alaska), L9 =500 nT. The GFZ website (https://www.gfz-potsdam.de/en/kp-index/) provides the 13 L9 values used for the Kp evaluation, showing values between 450 nT and 1500 nT; in particular, at Niemegk ([coordinates given], Germany) L9=500 nT."

We corrected the sentences as suggested by the reviewer (from line 46, revised ms). Canberra, the site contributing to Kp determination, is located in Australia.

**B3 - Lines 46 to 58 :** "For many years, *K* was manually derived by means of a conversion table containing the values of the maximum fluctuation A, expressed in units of nT, for each K value. With the introduction of digitized data and with the increasing access to computers, the manual estimation of K index was progressively substituted with automated algorithms and, nowadays, the reproducibility, one of the cornerstones of science, has become possible. (...) http://isgi.unistra.fr/softwares.php)."

Please, correct or amend the first sentence: "For many years, K was manually scaled by means of visual determination of the regular daily variation and of the consequent largest range of geomagnetic disturbances in the two horizontal components during a 3-hour UT interval. Then, K indices were determined by means of a conversion table between classes of ranges in nT and K indices."

Please, enclose and introduce the two following missing, but fundamental, references:

• Coles, R., and M. Menvielle (1991) - Some thoughts concerning new digital magnetic indices, Geophys. Trans., 36, 303-312.

• Menvielle, M. (1991) - Evaluation of algorithms for computer production of K indices, Geophys. Trans., 36, 313 -320.

Done. We adapted the sentences according to the referee's suggestions, and added the references: lines 51-58 in the revised manuscript.

**B4 - Lines 59 to 61:** "The International Real-time Magnetic Observatory Network (INTERMAGNET, http://www.intermagnet.org), of which IAGA is associated, endorses and recommends KASM for calculation of geomagnetic activity indices K according to the Adaptive Smoothed method (Nowozyński et al., 1991)."

Please, see comment A2. This sentence is incorrect, it has to be replaced by: "IAGA, through the ISGI international service, endorsed 4 different methods for calculation of local geomagnetic activity indices K. We used one of them the KASM method that used adaptive smoothed method (Nowozyński et al., 1991)."

Taking into account the previous statements, we modified the sentence suggested by the reviewer before including it in the revised ms, replacing the old one (lines 68-70 of the revised ms).

B5 - Line 63: "(...) the code derives daily values without fluctuations (mainly daily variation)."

Please correct the wording. "(...) the code estimates the regular daily variation." The sentence is modified according to the suggested one (lines 71-72 of the revised ms).

**B6** - Lines 65 to 70: "We want to point out that it does not exist an unique K9 at a given geomagnetic latitude since the geomagnetic activity shows a well known magnetic local time (MLT) dependency and, in addition, each site could be affected by different local features such as, for example, crustal anomalies (Chiappini et al., 2000) and/or coast effect (Parkinson, 1962; Regi et al., 2018). For the inclusion of a new geomagnetic observatory into the INTERMAGNET network, K9 should be assigned, for example, by comparing geomagnetic field variations between the new observatory and the historical ones for which K indices are estimated by using well defined K9 levels, obtained from a long time observation."

**This paragraph is entirely false.** It does exist a unique L9 at a given geomagnetic latitude. The 4 softwares endorsed by IAGA are taking care of the determination of the regular daily variations and are, by construction, considering the day-to-day variability. A simple plot of the regular daily variations extracted from softwares shows it clearly. Although one has to dig into the code and extract the relevant information. Indeed, codes available at ISGI are designed for operational purposes and were designed considering that the user knows their internal functioning.

At a particular magnetic observatory L9 is defined by the distance ? to "oval auroral" "auroral oval" modelled as the +/-69° latitude

CGM around 1965.

See:

• Mayaud, P.-N. (1968) - Indices Kn, Ks et Km, 1964-1967, Ed. C.N.R.S., Paris, 156 p. Especially the Figure A1 page 34.

A more recent paper is using that historical reference and may appear less 'arid' to the authors of the present paper:

• Lockwood, M., A. Chambodut, L. A. Barnard, M. J. Owens, E. Clarke, and V. Mendel. 'A Homogeneous Aa Index: 1. Secular Variation'. Journal of Space Weather and Space Climate 8 (2018): A53, doi: 10.1051/swsc/2018038.

Especially the Figure 3 page 6 and the related section 2.

Bartels defined the L9 of a particular magnetic observatory from hand-scaling of ranges and subsequent statistical study with the intent of producing a geomagnetic disturbance characterisation that does not depend significantly on the location of a sub-auroral, mid- or low- latitude observatory.

However, since beginning of the fifties, and even more since the digital era, the empirical method was put apart and the definition of L9 was chosen with regard to distance ? to "oval auroral" "auroral oval" modelled as the +/-69° latitude CGM. (One has to note that this method remained still unperfect, as the distance to the "oval auroral" "auroral oval" is changing with the main field but this is beyond the scope of your paper here.)

We believe that our paragraph is not false. Indeed, it does not exist a unique L9 at a given geomagnetic latitude for, at least, two main reasons: the crustal contribution to magnetic signals and the coast effect as the referee states at point D2 "...WNG comparison shows an underestimation of L9 values. Would it be possible that the location of WNG observatory nearby the shore (around 10 km to the North sea) leads to a possible bias in daily regular variation estimation in K indices calculation?"

We agree with the referee on the fact that the MLT dependency of magnetic disturbances can be smoothed along long time observations. This specification has been included in the revised ms at lines 76-77.

**B7 – line 79: "LMP is the southernmost observatory in Europe"**

Please, correct this statement which is false GUI (Guimar-Tenerife) magnetic observatory is southernmost, not speaking about French austral territories...

Guimar-Tenerife is located in African territory even if it politically belongs to Spain. The authors mean that LMP is the southernmost observatory in the European territory. We made it more clear in the text (line 89 in the revised ms).

**B8** – **lines 88 to 90:** "Our investigations suggest that NGK is the best reference observatory for Italian geomagnetic observatory of DUR, probably due to the closest magnetic local times: by comparing DUR with NGK we estimated a reliable DUR K9 level of 320 nT. Finally, by comparing also LMP with NGK, a reliable LMP K9 level of 310 nT is estimated."

A simple computing, considering distance to the "oval auroral" "auroral oval", leads to:

 for historical determination (without secular variation, Mayaud's method) L9 DUR = 356 nT and L9 LMP = 315 nT; • for January 2019 determination (taking the "oval auroral" "auroral oval" given by IGRF) L9 DUR = 354 nT and L9 LMP = 312 nT.

The results agree with the one provided by the authors. (The agreement is less striking for DUR as its geomagnetic latitude is beyond the range of possible K indices determination.) CGM coordinates of the Italian magnetic observatories remain quite constant along time as the  $L9_{NGK}$  does.

Please, correct or explain the part of the sentence saying that "probably due to the closest magnetic local times". If the L9 values are wisely chosen, then, local K indices statistical repartition along K values does not depend significantly on the location of the observatory. But in any case, a comparison along time obviously does show a clear Local Time (or Magnetic Local Time) dependence as the magnetic disturbances are impacting differently the day, dawn, dusk or midnight quarter (e.g. : K-derived magnetic indices in 4 Magnetic Local Time sectors ; see Chambodut, A., A. Marchaudon, M. Menvielle, F. ElLemdani and C. Lathuillere (2013) - The K-derived MLT sector geomagnetic indices, Geophys. Res. Lett., 40, 4808-4812, DOI:10.1002/grl.50947.)

The referee's suggestion helps us to consolidate our conclusion that NGK is the best reference observatory for DUR (lines 99-100 and lines 261-263 revised ms and Chambodut's reference included). We also mentioned the L9 values leaded by Mayaud's method (lines 265-267 in the revised ms).

**section 2-"Data and methods of analysis"**

**C1-** Please, which time-resolution magnetic observatory data are you using with the KASM method? Please indicate it. A first guess would be "minute data computed from second data using INTERMAGNET 1s to 1min filter".

We added the specific note given by the referee (line 108 of the revised ms).

**section 3-"Experimental results" subsection 3.1 "K9 empirical estimation"**

**D1- regarding Figure 3 and the related description in the section.** The discrepancies observed are for low K indices. Do the authors have an explanation? Can it be a limitation of the K index derivation scheme in really quiet and quiet magnetic conditions?.

**We made more clear the description of figure 3 (lines 147-149 revised ms)**

**D2- regarding Figure 4 and the related description in the section 2.** The discrepancies observed between the black and red curves are, for both compared observatories (LMP and DUR), in the same direction. WNG comparison shows an underestimation of L9 values. Would it be possible that the location of WNG observatory nearby the shore (around 10 km to the North sea) leads to a possible bias in daily regular variation estimation in K indices calculation?

K determination at WNG could be affected by geomagnetic coast effect; we added this statement shortly in the revised ms (lines 157-158)

**D3- lines 146 to 150:** "In addition, the higher correlations are obtained by using NGK, probably due to the lower latitude (i.e. closer to the Italian observatories) and the closer MLT with respect to DUR (table 1). Also at LMP, even if the MLT is closest to that of WNG, the higher correlation is found with NGK: this result suggests that latitudinal effects are dominant with respect to MLT ones. This can be well understood taking into account that the MLT range of all selected observatories is within 11 minutes, well shorter than the 3-hour interval used for K determination." Please, consider D1 and D2 questions.

Yes, we already made considerations in the ms, according to D1 and D2 questions

**D4-lines 179 to 180:** "We point out how the distributions are close to each other, suggesting that FMI and KASM are consistent algorithms, (...)"

You obtained the same results as: Coles & Menvielle (1991), Menvielle (1991) and Menvielle et al. (1995).

We mention this aspect since it reinforces our results, adding the suggested citations (line 195 of the revised ms).

**D5- regarding Figure 5 and the related description in the section 2:**

Please explain why Figure 5 (left) has to be symmetric for a better L9 value considering that only 2 years of data are used, a tiny part of the solar cycle?

We believe that only a long time series could lead to a true symmetric pattern. Any asymmetric pattern indicates that L9 is underestimated or overestimated. Our statistic is not so wide but the L9 value corresponding to the maximum correlation also gives the maximum symmetry. We are also confident that K index derived by L9 here estimated is based to the NGK K index which are calibrated for long time series, including more than one solar cycle. This concept is already included in the original version of the ms.

**D6- regarding Figure 7 and the related description in section 2:** The authors are in fact here doing a comparison of K derivation softwares. The discrepancies observed are similar with the ones observed between Asm method and FMI method in Menvielle et al. (1995).

We added the suggested citation.

subsection 3.2-" Comparison with a previous K9 estimation method"

**E1- lines 190 and 191:** "According to Mayaud (1980), an approximate value of  $\delta$  could be given by  $\delta$ =69°- $\lambda$  but this is really just a rough approximation."

Please, do not be so rude. Mayaud method is still the one in use that proved, and still proves, its robustness. The results of the present paper are similar to the ones obtained with Mayaud's method, see B8. Furthermore, the correct reference here is Mayaud, P.-N. (1968) - Indices Kn, Ks et Km, 1964-1967, Ed. C.N.R.S., Paris, 156p. (available at http://isgi.unistra.fr/Documents/Books/Mayaud CNRS 1968 complete.pdf).

The authors just commented the definition of  $\delta$  that can be approximated by  $\delta$ =69°- $\lambda$  as suggested in Mayaud (1980). We rewrite the sentence (line 206 in the revised manuscript). We also added the suggested Mayaud(1968) reference at lines 201 and 205.

**E2-lines 193 to 200:**

Please correct the approximated equation you are using. Mayaud is using a 4th degree polynomial.

Mayaud 1968 approximated the  $L/L_0 - \delta$  relationship (Fig. A1) by a combination of two hyperbola. In our manuscript we simply used the L and  $\delta$  values reported in Table 5 by Mayaud (1980). By choosing the new variable x=1/ $\delta$  we obtained that a linear fit well reproduces the L9-x relationship, as it can be seen in Fig. 8 of our manuscript. Therefore, replacing K9 with L9 according with M2 comment, we rewrite the equation in the revised ms as  $L9(x) = \alpha x + \beta$ .

**E3 lines 192 to 231:**

This part of the present paper is largely incorrect. Please, read Mayaud (1968) (or Lockwood et al. (2018), page 5 to 7, for a more "modern english" explanation) and correct.

As stated above, we based this part of our analysis on the  $\delta$  and L values provided by Mayaud (1980) and following our observation that L9(x) relationship is linear (Fig. 8). Therefore, we are confident that our analysis is correct. Indeed, as stated by the reviewer, our results agree with the results given by the well-known method endorsed by IAGA and implemented by ISGI.

**section 4-" Discussion and Conclusions"**

**F1-lines 234 to 236:** "The modern automatic procedures for calculating local K index values, with the setting of some a-priori criteria, have to be carefully verified for their permanent validation in terms of accuracy and stability when delivered to the scientific community."

Please erase this sentence. The present paper has to be reviewed with major revisions. Done.

F2- line 237: "(...) This code is distributed by the INTERMAGNET consortium(...)" Please see comments A2 and B4.
We rewrite the sentence accordingly with comments A2 and B4 (lines 248-252 in the revised version of the ms).

**F3- lines 239 to 241:** "(...) the K9 value, which represents the minimum value of the amplitude extent in the H component of Earth's magnetic field when the local K value reaches the integer 9, the highest level in a scale which ranges from 0 to 9."

Please correct: "L9 value, the so-called "K=9 lower limit" allows to determine, for each magnetic observatory, the conversion table between classes of ranges and K indices."

Done: lines 253-255 in the revised manuscript.

**F4- lines 263 to 272:** "Moreover, Mayaud (1980) note that the limitation of the method they propose is that it is conceived for sub-auroral and mid latitudes; indeed, they suggest that for lower latitudes a constant K9=300 nT can be chosen. This very approximate value is not very far from the values we estimate (320 nT for DUR and 310 nT for LMP), but would certainly be not accurate as them in the comparison with the values from other reference observatories: indeed our results clearly show that a very precise K9 limit is necessary for obtaining K values well consistent at different sites. As a final remark, from the overall view of this work, we are also definitely convinced that the habit to round the value of K9 in multiples of 50 nT is a simplified approximation, firstly suggested by Bartels et al. (1939), a practice that needs to be abandoned. This approximation is still adopted in some cases, demonstrating that perhaps a critical revision has not been applied yet, differently from the case of Kakioka observatory (Japan) where K9 has a convincing value of 296 nT."

Please erase that part. Mayaud (1968) clearly stated that the magnetic observatories towards polar areas or towards equatorial regions ( $58^{\circ} > |CGM |$  latitude $| > 29^{\circ}$ ) are under magnetic conditions (e.g.: field aligned currents, magnetospheric ring current, etc) that do not allow to produce K indices comparable to mid-latitude ones. The Figure A1 of Mayaud (1968) clearly show a hyperbola with two asymptotes. One may calculate K indices for sub-equatorial or sub-polar geomagnetic observatories but without real physical meaning. The activity of the magnetic field in these observatories cannot be assessed with the proposed softwares. This is also the reason why K-derived Magnetic indices, such as Kp, aa or am, are only fully meaningful for midlatitude. Dst (equatorial) or PC (polar) are not K-derived indices.

Please, do not patronise and give recommendation. Indeed, the L9 values are presented onto the ISGI website for each magnetic observatory. The rounding of L9 values may easily be overcome by the use of the well-know FKC table developed and used by both Bartels and Mayaud. Magnetic Observatories do not need to update or change their L9 values. Homogeneity of the series is of primary importance.

For each magnetic observatory, the only mandatory point and message that should be given to the whole community is: Please, provide carefully in the metadata the L9 value that was actually used for K indices calculation.

We agree with the reviewer that magnetic observatories do not have to change their L9 values since the homogeneity of the series is of primary importance. In the revised manuscript we changed lines 289-293 according with the reviewer's suggestions.

Regarding the latitude of our observatories (AACGM latitudes 27.9° N and 35.9° N for LMP and DUR respectively) we point out that they are within the range of validity for "s" scaling published by Mayaud (1968), i.e. 29-58°. Although LMP is slightly outside this range, the L9 here computed by our method is consistent with that indicated by a ISGI member (personal communication).

**Replies to Reviewer #3**

The Authors thank the reviewer for her/his helpful comments which increased the quality of our work. Here below we reported the reviewer's comments (in black) and our replies (in red).

**Major questions**

a) The paper presents the situation as if each observatory has to choose its own k9. But, according to the International Service of Geomagnetic Indices (ISGI), the ISGI headquarters are in charge of the computation of k9 value for each magnetic observatory. ISG is a service of the International Association of Geomagnetism and Aeronomy (IAGA) which recognized "the unique role of the International Service of Geomagnetic Indices (ISGI) in the derivation, publication, and dissemination of these indices" (IAGA, Resolution No. 9 (1989). So, this paper should present its conclusions as an academic effort to check the assigned k9 values rather than to propose a new operational value.

We tried to determine L9 values (note: In the whole manuscript we substituted "K9" with "L9" as suggested by #2 reviewer) by means of a statistical approach and comparison with historical reference observatories. The values we found are in good agreement with the ones directly obtained from Mayaud's method (personal communication, by ISGI members). We are convinced that the contribution of ISGI in providing the L9 value is fundamental and that L9 values of observatories which have been providing the K index in the past (often rounded in multiple of 50 nT) cannot be changed since homogeneity of the series is of primary importance. However, before a new observatory starts providing K index values, it is worth evaluating K indices with a provisional L9 value, assigned by ISGI, and then refine it with a procedure like the one we have shown, based on the comparison with reference, well set, historical observatories. The authors believe that discussions derived from this deal help to reason about the suitability of past procedures with respect to the modern capability of automatic computations. In the revised version of the manuscript these considerations are included in lines 264-266 and 288-292.

b) K index is a coarse indicator of magnetic activity which simplifies the environmental situation into 10 digits for the sake of having a simple way to classify a magnetic disturbance. The scale is not linear nor logarithmic but a sort of personal choice of Bartels to have some events in each interval. Specially, in the high numbers, very diverse disturbances are assigned to the same figure (e.g. k=8 in Nimeck could be 331 nT or 500 nT) Moreover, Sq estimation, necessary because it should be removed before k computation, is rather subjective and involve a large uncertainty. In each algorithm, Sq is interpreted differently (Menvielle et al., 1995). The authors claim the necessity of giving k9 values in units of nT or even with tenths of nT (Ln 268-273) but this it would be misinterpreted as if they were very precise when they are not!

The authors agree with the question arose by the referee on the pertinence of having a precise (even on the order of unit of nT) value of K9 when the procedure of obtaining such value has a relatively large margin of subjectivity. Nevertheless, in the era of automatic computation, if a determination of a number must be done, the authors think that the final number should be used as it results from computation or at least rounded at the closest tenth of nT. Indeed, in our statistical analysis a 10 nT resolution for L9 is chosen.

**Minor questions**

c) Niemek (NGK) was the reference observatory where this scale was defined by Bartels. The rest of other observatories where assimilate to this to create distributions similar to that one. So, the comparison of the Italian observatories with this observatory has more sense than with other German observatory (WNG). In fact, this observatory although being located very close to NGK, has not a perfect correspondence in distribution with (19% deltaK=-1, ln 130).

The referee correctly interpreted the reason why the authors made use of Niemek (NGK) observatory for comparison with the Italian geomagnetic observatories. Nevertheless, for a further comparison, we also used WNG demonstrating that the best correlation is obtained with NGK.

d) However, as K index measures the effect of auroral activity, it seems more reasonable to compare Italian observatories' distribution with other observatories with similar latitude. Moreover, it is well known that k9 limit does not follow a regular law with the angular distance to the auroral zone (Mayaud, 1980).

Once again the referee well interpreted the limits highlighted by the authors when comparisons among observatories located at different latitudes are used. On the other hand, the amplitude of magnetic disturbances has a dependence on (magnetic) local time which affects the K index values (Chambodut et al. (2013)). Since there are not historical observatories located at a similar latitude and, at the same time, not too far in local time, we preferred to use NGK.

e) Although it is true that digital algorithms grant reproducibility (Pg 2 ln 49), this does not mean that they are more certain. In the past, K index was "estimated" for manual procedures; but, now, an automatic algorithm also produces "estimated" values. And, of course, different algorithms would produce different values (pg9 ln255).

f) Comparing the distribution of Italian indices generated by KASM algorithm with NGK and WNG indices generated by FMI algorithm (Ln 137) is a rough way to do this because these distributions change with the algorithm and even with the year being considered (Figure 7).

**Points e) and f)**

We modified the manuscript introduction at lines 50-57, also according to #2 reviewer's suggestions. Different algorithms for the k index "estimation" produce different values. That's exactly what comes from the ms, simply comparing the distribution of K indices generated by KASM and FMI algorithms, which is probably a granted result but not so universally known. However, our results (Fig.7) show that FMI and KASM are very similar. Indeed, both algorithms are among the 4 endorsed and recommended by IAGA (through ISGI). Maybe an open discussion at geomagnetic community level could be useful to establish new procedures or to revise the old ones, with more critical and fruitful approach.

h) The correlation analysis used to obtain the best value of k9 (figure 4) presents a flat and asymmetric shape in an interval ranking for more than 50 nT. There, any change, as the use of a different algorithm, would produce a different maximum. So, I would not take the new values as a step forward. In fact, final results (those choosing a new value of k9 for DUR (fig. 5) implies a variation of 10% of population in deltaK=-/+1, in the limit of the precision of the method.

The authors show that the correlation analysis allowed to obtain a new L9 value for Duronia observatory, slightly improving the global performance of its K index which depends on the appropriate L9 values. For this reason the authors believe that, even in the limit of the precision of the method and of the comparison proposed in the ms, the final results could be a useful improvement for DUR observatory, as well as for LMP which has to start the K index computation, and a stimulating hint for many scientists involved in the work of geomagnetic observatories.

**On the validation of K index values at Italian geomagnetic observatories**

Mauro Regi1, Paolo Bagiacchi2, Domenico Di Mauro2, Stefania Lepidi1, Lili Cafarella2

1Istituto Nazionale di Geofisica e Vulcanologia, 67100, L'Aquila, Italy

5 2Istituto Nazionale di Geofisica e Vulcanologia, 00143, Rome, Italy

Correspondence to: Mauro Regi (mauro.regi@ingv.it)

**Abstract.** Local K index and the consequent global Kp index are well established three-hour range indices used to characterize the geomagnetic activity. K index is one of the parameters which INTERMAGNET observatories can provide and it's widely used since several decades, although many other activity indices have been proposed in the meanwhile. The

- 10 method for determining the K values has to be the same for all observatories. INTERMAGNET consortium recommends the use of one of the 4 methods endorsed by the International Service of Geomagnetic Indices (ISGI) in close cooperation and agreement with the ad-hoc working group of International Association of Geomagnetism and Aeronomy (IAGA). INTERMAGNET provides the software code KASM, designed for an automatic calculation of K index according to the Adaptive Smoothed method. K values should be independent on the local dynamic response, therefore for their
- 15 determination each observatory has its own specific scale regulated by the L9 lower limit, which represents the main input parameter for KASM. The determination of an appropriate L9 value for any geomagnetic observatory is then fundamental. In this work we statistically analyze the K values estimated by means of KASM code for the Italian geomagnetic observatories of Duronia (corrected geomagnetic latitude  $\lambda \sim 36^{\circ}$  N) and Lampedusa ( $\lambda \sim 28^{\circ}$  N) comparing them with the German observatories of Wingst and Niemegk. Our comparative analysis is finalized to establish the best estimation of the
- 20 L9 lower limit for these two stations. A comparison of L9 lower limits found for the Italian observatories with results from a previous empirical method was also applied and used to verify the consistency and reliability of our outcomes.

**1** Introduction**

- In their pioneering work, Bartels et al. (1939) introduced the three-hour-range K index with the purpose of quantifying the solar wind (or particle) effects on the geomagnetic field. K index is represented with an integer in the range 0–9 ("K" is from the German word Kennziffer meaning "characteristic digit") with 0 and 1 being an indication of quiet condition and 5 or more referring to an increased level of magnetic activity, generally related to a geomagnetic storm. It is derived for a specific observatory from the maximum fluctuations of horizontal components observed on a magnetogram during a three-hour interval, evaluated as difference between maximum positive and negative deviations with respect to a reference curve which
  essentially reflects the local diurnal variation at the observatory. These maximum deviations may occur at any time during
  - the 3 hour period. The proposed K index was originally calculated for Niemegk observatory.

As a natural consequence of K index, the planetary geomagnetic activity index Kp was proposed by Bartels (1949). It is derived from the standardized K index (Ks) of 13 magnetic observatories at mid latitude and it is representative of the large spatial-scale of the solar wind-magnetosphere coupling energy. Therefore, K index is the fundamental parameter for Kp

- 35 estimation; Kp, as any other index, has limitations and drawbacks; however, it's precious since it's an historical parameter and long data series are available; it is widely used, for example, in space weather applications, for identify quietest days (Johnston, 1943) used also in the IGRF modeling, for verifying solar wind driven modulation in the atmospheric parameters during disturbed conditions (Regi et al., 2017).
- The main difficulty for K indices evaluation is to assign a proper quasi-logarithmic scale to the geomagnetic fluctuations that 40 satisfy the principle of the assimilation of frequency distributions (AFD): the frequency distributions (or occurrences) of K index values at different sites are, in principle, the same (Bartels et al., 1939). In other words, A values vary from observatory to observatory in such a way that the historical rate of occurrence of certain levels of K is about the same at all observatories (Bartels-Mayaud rules). This implies that, for a given K value,  $A_{\rm K}$  increases with increasing latitude, and the fundamental quantity for the K index calculation is represented by the minimum amplitude L9 corresponding to K=9, from
- 45 which also the other  $A_{\rm K}$  values are derived.

50

For Kp determination (Bartels et al., 1939), from higher to lower latitude, at Sitka (AACGM latitude  $\lambda \sim 60^{\circ}$  N, Alaska, Alaska), L9 =1000 nT while Canberra (AACGM latitude  $\lambda \sim 45^{\circ}$  S, Australia), L9 =500 nT. The GFZ website (https://www.gfz-potsdam.de/en/kp-index/) provides the 13 L9 values used for the Kp evaluation, showing values between 450 nT and 1500 nT; in particular, at Niemegk (AACGM latitude  $\lambda \sim 48^{\circ}$  N, Germany) L9=500 nT. In the present manuscript all the AACGM coordinates are computed by using the Shepherd (2014) algorithm, applied to year 2017.

The original determination of K indices (Bartels et al., 1939) required hand scaling of analogic magnetograms. For many years, K index was in fact manually scaled by visual determination and removal of the regular daily variation; the remaining largest amplitude of geomagnetic disturbances in the two horizontal components during each 3-hour UT interval, was used to determine the K index values from a conversion table between classes of ranges in nT and K indices.

55 The question of the derivation of geomagnetic indices from digital data arose at the end of the seventies of the last century. Different algorithms enabling computer derivation of K indices were then developed and carefully assessed in the frame of an international comparison organized by the IAGA Working Group "Geomagnetic indices" (Coles and Menvielle, 1991; Menvielle, 1991; Menvielle et al., 1995).

This implies the production of computer plots of digital data with scale values similar to those of photographic

60 magnetograms (Menvielle et al., 1995). The International Association of Geomagnetism and Aeronomy (IAGA, http://www.iaga-aiga.org/) promotes tools or methods able to make it easier to keep track of files and analyses done on computers.

Different methods were proposed and carefully compared and assessed in occasion of an international meetings organized by the IAGA Working Group "Geomagnetic indices" during the Vienna IUGG general Assembly in 1991 and four methods

65 were acknowledged: FMI (provided by Finnish Meteorological Institute, Finland), LRNS (Hermanus Magnetic Observatory, CISR, South Africa), KASM (Institute of Geophysics, Polish Academy of Science) and USGS (USGS, USA), whose Fortran 77 codes are available at the International Service of Geomagnetic Indices (ISGI, http://isgi.unistra.fr/softwares.php).

We used one of the 4 methods endorsed by IAGA, through the ISGI international service, for calculation of local geomagnetic activity indices K and, in particular, the KASM method that is based on adaptive smoothed method
70 (Nowozyński et al., 1991). For the calculation of the K index, IAGA formatted files are used by KASM code. It requires three daily files, the one under analysis and the files of the previous and following days on which the code estimates the

regular daily variation. The code also needs as input parameters the L9 value and the yearly average of the H component relative to the year of interest.

We want to point out that it does not exist a unique L9 at a given geomagnetic latitude since each site maight be affected by 75 different local features such as, for example, crustal anomalies (Chiappini et al., 2000) and/or coast effect (Parkinson, 1962; Regi et al., 2018). Moreover, there is the inevitable MLT dependencies of magnetic disturbances which can be smoothed out trough statistical approach, considering long time observations. For the inclusion of a new geomagnetic observatory into the

INTERMAGNET network, an L9 value can be initially assigned according to the ISGI indication but it can be refined by comparing long term geomagnetic field variations at the new observatory and at historical ones for which K indices are
 estimated by using well defined L9 levels.

We used the geomagnetic data from the two Italian geomagnetic observatories at Duronia (DUR) and Lampedusa (LMP), evaluating the K index with the purpose of estimating the best L9 value for each observatory.

DUR observatory is operating in Central Italy in the area of the village of Duronia (geogr. coordinates: 41°39'N, 14°28'E, 910 m a.s.l.). It was installed at the end of 2007 in the framework of MEM (Magnetic and Electric fields Monitoring) Project

85 that aims to investigate the environmental electromagnetic signals in the ULF-VLF (0.001 Hz - 100 kHz) frequency band, and was granted as geomagnetic observatory in 2012, when it was included in the INTERMAGNET network (http://www.intermagnet.org), replacing the historical geomagnetic observatory at L'Aquila partially damaged after the local Mw 6.2 earthquake in 2009.

LMP is the southernmost observatory in European territory (geogr. coordinates: 35°31'N, 12°32'E); it was installed in 2005 90 and is regularly working since 2007.

Up to now, K index was evaluated only for DUR observatory, using L9=350 nT, both for hand-scaling (since 2012) and for KASM program (since 2017).

In this work we evaluated L9 throughout a correlation analysis performed between K index at DUR with that provided by historical observatories. In order to take into account the magnetic local time dependency of reference K index, European

95 observatories were selected. As possible reference observatories we chose Wingst (WNG) and Niemegk (NGK), since they are among the 13 observatories that contribute to the Kp estimation and their local magnetic time is quite close to that of our Italian observatories.

Our investigations suggest that NGK is the best reference observatory for Italian geomagnetic observatory of DUR, probably due to the closest magnetic local times: indeed the amplitude of magnetic disturbances has dependence on (magnetic) local time which affects the K index values (Chambodut et al. (2013)). By comparing DUR with NGK we estimated a reliable

100 time which affects the K index values (Chambodut et al. (2013)). By comparing DUR with NGK we estimated a reliab DUR L9 level of 320 nT. Finally, by comparing also LMP with NGK, a reliable LMP L9 level of 310 nT is estimated.

**2 Data and methods of analysis**

Geomagnetic field variations at Italian geomagnetic observatories of DUR and LMP are measured by using three-axis
fluxgate magnetometers along the northward (*H*), eastward (*D*), and vertically downward (*Z*) directions in the geomagnetic reference frame at 1 s sampling rate. Following the INTERMAGNET directives, geomagnetic time series are also stored as daily archives at 1 min sampling rate, according to the IAGA 2002 format.

In this work we used minute data computed from second data using INTERMAGNET 1s to 1min filter, available in the time interval 1 January 2017 - 31 December 2018, a temporal window which falls in the lower part of the sunspot number curve for the cycle 24 (Upton & Hathaway, 2018).

These data are used for estimating K indices by using KASM algorithm which is recommended by INTERMAGNET. In this

work, the definitive L9 level at DUR is empirically estimated throughout the following procedure:

- a) we selected a reference observatory;
- b) K index time series at DUR are computed by using KASM for different L9 values (KL9) in the range 200-400 nT with a step size of 10 nT;
- c) each KL9 index time series at DUR is compared with K index time series at reference observatory through correlation analysis;
- d) the definitive L9 level at DUR is estimated in correspondence of the maximum correlation coefficient.

[revised manuscript text omitted]

agreement with Coles and Menvielle (1991), Menvielle (1991) and Menvielle et al. (1995).

**3.2 Comparison with a previous L9 estimation method**

As explained in the introduction, the geomagnetic indices are historically assigned throughout visual inspection of magnetograms. The main difficulty for K indices evaluation is to assign a proper value for the L9 limit from which determining the quasi-logarithmic scale to the geomagnetic fluctuations in order to satisfy the AFD principle (Bartels et al., 1939). Mayaud (1968; 1980) proposed a method for calculating the geomagnetic activity level *L* at a given site by comparing the amplitude of geomagnetic fluctuations at the reference observatory (*A*0) with that, for example, at new one (*A*) as follows: L=L0*A*/*A*0, where *L*0 represents the level of geomagnetic activity at the reference observatory, equivalent to L9, and all quantities are dependent on δ= min[ λoval- λ ], i.e. the minimum angular separation between the site, located at geomagnetic latitude λ, and the auroral region, at λoval. According to Mayaud (1968; 1980), an approximate value of δ could be given by δ=69°-λ, where the latitude 69° in the corrected geomagnetic coordinate system defines the auroral zone.

We searched a simple relationship which relates L9 (or *L*) to the geomagnetic latitude of the observatory.

As showed by Mayaud (1980), L9 increases with decreasing δ (L9∝δ-1), as expected for a geomagnetic field induced by a current system. Figure 8 shows L9(δ) (blue points) provided in Tab. 5 by Mayaud (1980), considering only northern
hemisphere. These points are well represented by a linear law considering an increasing induction effect with increasing parameter x=1/δ.

Therefore, by using x=1/ $\delta$ , previous relationship is linearized and can be formulated as follows

$$L9(x) = \alpha x + \beta . \tag{1}$$

**215 The results of the linear regression analysis performed on the experimental points are also reported in Fig.8.**

Equation 1 is therefore useful for estimating a reasonable L9 limit at a different site. In order to evaluate L9 at DUR, LMP and, for a comparison, at NGK observatories, the corresponding  $\delta$  parameter is required. However, it is not clear how  $\delta$  was estimated by Mayaud (1980), since it requires, for example, an auroral oval model for estimating the  $\lambda_{oval}$ , and an IGRF model for evaluating the geomagnetic latitude  $\lambda$  of a given site (this aspect will be further discussed at the end of this section).

In this regard, we empirically estimated  $\delta(\lambda)$  by a linear fit of the experimental data reported by Mayaud (1980). Figure 9 shows experimental points (red stars) and the corresponding linear law (red line)

$$\delta(\lambda) = a\lambda + b , \qquad (2)$$

225

220

which allows us to extrapolate an estimation of the theoretical  $\delta_{th}(\lambda)$  for the observatories of DUR, LMP and, for comparison at NGK too (blue circles), where  $\lambda$  represents the corrected geomagnetic latitude used by Mayaud (1980). Finally, by inserting  $\delta_{th}$  into the Eq. (1) we estimated the L9( $\delta_{th}$ ) level at the observatories of DUR, LMP and NGK.

All these results are reported in Tab.2, which also shows for a comparison, L9 obtained by computing  $\delta_a=69^{\circ}-\lambda$ , and L9exp 230 experimentally derived by our calibration procedure (L9exp), together with the 95% confidence intervals for the fitted L9 values.

It can be seen that all  $L9_{exp}$  are consistent to each other within their respective confidence interval, at a given observatory. The small difference between  $L9_{exp}$  and  $L9_{th}$  could be due to a different method for calculating the geomagnetic coordinates used in Mayaud and in this work (we use AACGM). In order to verify this hypothesis, we performed a correction on the key parameter  $\delta(\lambda)$  as follows:

235 parameter  $\delta(\lambda)$  as follows:

we computed the AACGM latitudes  $\Lambda$  of geomagnetic observatories from Tab. 5 of Mayaud and corrected  $\lambda$  through linear relationship  $\lambda_C = l \Lambda + m$ ;

we performed a linear fit of  $\delta(\lambda)$ ,  $\lambda_c$ , which provides the relationship for the adjusted  $\delta_A(\lambda_c)$ ;

finally we performed a linear fit of L9( $\delta$ ),  $\delta_4$  which provides the adjusted L9A( $\Lambda$ ) estimated at our geomagnetic observatories as a function of AACGM latitude.

With respect to the L9( $\delta$ ) it can be seen that the adjusted L9A( $\Lambda$ ) (shown in Tab.2) are closer to the experimental L9exp, indicating that the correction on geomagnetic coordinate makes a significant contribution on the L9 estimation. LMP is the only one that shows a discrepancy between  $L_{9exp}$  and  $L_{9}$  here estimated with different methods. A possible reason of this discrepancy lies in the low latitude of LMP observatory where the ring current and/or electrojet currents dynamics could affect L9 estimations.

245

240

255

260

**4 Discussion and Conclusions**

Four different automated methods for calculation of local geomagnetic activity indices K were endorsed by IAGA, through the ISGI international service, and distributed from ISGI (http://isgi.unistra.fr/softwares.php) web site. For the Italian 250 geomagnetic observatories of Duronia (DUR) and Lampedusa (LMP) we used one of them, i.e. the KASM algorithm that is based on adaptive smoothed method (Nowozyński et al., 1991) which is the only one provided also by INTERMAGNET (https://www.intermagnet.org/publication-software/software-eng.php)

An input parameter required by KASM code, as well as FMI code, is the L9 value, the so-called "K=9 lower limit", which allows to determine, for each magnetic observatory, the conversion table between classes of geomagnetic field variation ranges and K index values.

We found L9 values for DUR and LMP through a correlation analysis using as reference the corresponding data from the two European observatories of Wingst (WNG) and Niemegk (NGK), both located in Germany. The choice of these two observatories was prompted by the fact that they are among the 13 observatories which provide their K indices for the determination of the planetary Kp index and moreover, their magnetic local time is very close to that of the Italian observatories.

We note that NGK is the best reference observatory for Italian geomagnetic observatory of DUR, possibly due to the closest magnetic local time; indeed the amplitude of magnetic disturbances has dependence on (magnetic) local time which inevitably reflects on different K index values (Chambodut et al. (2013)).

Based on a dataset related to a couple of years (2017 and 2018), this analysis allowed to establish that for DUR and LMP the

265 L9 values are 320 nT and 310 nT, respectively. These values are in good agreement with the ones directly obtained from Mayaud's method which leads to approximately 355 nT and 315 nT for DUR ans LMP, respectively (personal communication, by ISGI members). The method can be generalized and applied to every observatory in the world to verify if the choice to scale local fluctuations of the Earth's magnetic field is properly calibrated by a suitably selected L9 value, regardless if manually or automatically computed. Our analysis also highlighted the possibility of establishing a linear

270 relationship between a pair of analyzed observatory datasets that can be useful for predicting or deriving the index of one when the other is known.

Another interesting result that we found is related to the consistency of the KASM code and the FMI code, the latter in use at the two German observatories for the K index computation and subsequent release. Although FMI code is based on a different procedure, we verified that the results obtained are consistent with those obtained by KASM code and stable in the two-year time interval, although with a slightly different value of the input L9 parameter. This confirms that the choice of a

certain algorithm in place of another does not invalidate the results.

Before the introduction of automatic procedures, based on the definition introduced by Bartels et al. (1939) for the K index concept, in the '80s of the last century Mayaud (1980) used an empirical relation to calculate the level of the local magnetic activity L (equivalent to the L9 values) for a generic point of observation with respect to a referenced observatory. Through a
linearization process, we used this relation, which includes some approximations and the necessity of determining the minimum angular separation between the observational point and the auroral region, i.e. a method for determining the geomagnetic latitude, obtaining an independent estimate of the L9 values for our observatories which is consistent, within the 95% interval of confidence, with that obtained by our previous analysis. Moreover, Mayaud (1980) notes that the limitation of the method he proposes is that it is conceived for sub-auroral and mid latitudes; indeed, he suggests that for

- 285 lower latitudes a constant L9=300 nT can be chosen. This very approximate value is not very far from the values we estimate (320 nT for DUR and 310 nT for LMP), but would certainly be not accurate as them in the comparison with the values from other reference observatories: indeed our results clearly show that a very precise L9 limit is necessary for obtaining K values well consistent at different sites. As a final remark, from the overall view of this work, we are also convinced that the habit to round the value of L9 in multiples of 50 nT, firstly suggested by Bartels et al. (1939) cannot be changed for observatories
- 290 which have been providing the K index in the past since homogeneity of the series is of primary importance. However, before a new observatory starts providing K index values, it is worth evaluating K indices with a provisional L9 value, assigned by ISGI, and then refine with a procedure like the one we have shown, based on the comparison with reference, well set, observatories.

**295 Author contribution**

275

DDM, SL and MR planned the study. MR performed the data analysis and wrote the codes. PB studied the KASM code functionality and, together with MR, tested and validated its results. DDM, LC and SL improved the quality of the manuscript. All authors read and approved the final manuscript.

**300 Competing interests**

The authors declare that they have no competing interests.

**Acknowledgments**

The results presented in this paper rely on data collected at magnetic observatories. We thank the national institutes that 305 support them and INTERMAGNET for promoting high standards of magnetic observatory practice (www.intermagnet.org). We also thank Jürgen Matzka from Helmholtz Centre Potsdam GFZ German Research Centre for Geosciences (Germany) for providing K indices at Niemegk and Wingst.

Figure 1: European geomagnetic observatories used in this work